# Fusion of the Cas9 endonuclease and the VirD2 relaxase facilitates homology-directed repair for precise genome engineering in rice

Zahir Ali 1,4, Ashwag Shami1,2,4, Khalid Sedeek1,4, Radwa Kamel1, Abdulrahman Alhabsi 1,
Muhammad Tehseen3, Norhan Hassan1, Haroon Butt1, Ahad Kababji1, Samir M. Hamdan 3 &
Magdy M. Mahfouz1*

Precise genome editing by systems such as clustered regularly interspaced short palindromic repeats (CRISPR)/CRISPR-associated protein 9 (Cas9) requires high-efficiency homology-directed repair (HDR). Different technologies have been developed to improve HDR but with limited success. Here, we generated a fusion between the Cas9 endonuclease and the *Agrobacterium* VirD2 relaxase (Cas9-VirD2). This chimeric protein combines the functions of Cas9, which produces targeted and specific DNA double-strand breaks (DSBs), and the VirD2 relaxase, which brings the repair template in close proximity to the DSBs, to facilitate HDR. We successfully employed our Cas9-VirD2 system for precise *ACETOLACTATE SYN-THASE* (*OsALS*) allele modification to generate herbicide-resistant rice (*Oryza sativa*) plants, *CAROTENOID CLEAVAGE DIOXYGENASE-7* (*OsCCD7*) to engineer plant architecture, and generate in-frame fusions with the HA epitope at *HISTONE DEACETYLASE* (*OsHDT*) locus. The Cas9-VirD2 system expands our ability to improve agriculturally important traits in crops and opens new possibilities for precision genome engineering across diverse eukaryotic species.

[1] Laboratory for Genome Engineering and Synthetic Biology, Division of Biological Sciences, 4700 King Abdullah University of Science and Technology, Thuwal 23955-6900, Saudi Arabia. [2] College of Science, Biology Department, Kingdom of Saudi Arabia (KSA), Princess Nourah bint Abdulrahman University, Riyadh, Saudi Arabia. [3] Laboratory of DNA Replication and Recombination, Biological and Environmental Sciences and Engineering Division, King Abdullah University of Science and Technology (KAUST), Thuwal 23955-6900, Saudi Arabia. [4] These authors contributed equally: Zahir Ali, Ashwag Shami, Khalid Sedeek. *email: magdy.mahfouz@kaust.edu.sa

The CRISPR/Cas9 system has been customized for targeted genome editing, regulation of gene expression, and viral interference in diverse eukaryotic species, including plants[1–4]. In the *Streptococcus pyogenes* CRISPR/Cas9 system, a synthetic single guide RNA (sgRNA) directs the Cas9 endonuclease to a 20-nucleotide complementary sequence on the target DNA where, upon the recognition of the protospacer-associated motif (PAM, NGG), Cas9 makes a double-stranded break (DSB)[3,5,6]. This DSB can be repaired by the nonhomologous end joining (NHEJ) repair pathway, which produces imprecise repairs, or by homology-directed repair (HDR), which produces precise repairs based on a nucleic acid template[7,8]. The CRISPR/Cas9 system is widely used for gene editing in plants because of its ability to produce site-specific gene knockout (loss of function) or new functional variants of an allele via short insertions and deletions (Indels) from NHEJ repair[9,10]. Base editors can also produce single-base conversions in a short window of a target sequence[11,12]. However, improving specific agronomic traits of crop plants requires broad and precise control of the modifications produced by genome engineering.

HDR can produce precise genome edits, ranging from a single-base replacement to insertions of multiple kilobase-long fragments, such as a specific motif or promoter, at user-defined loci[13]. The recent application of the CRISPR/Cas9 system with HDR for targeted improvements in many eukaryotic species, including plants, has revolutionized our ability to engineer genome sequences[14,15]. However, HDR is less efficient in plants than in animals, and several factors limit efforts to use HDR for plant genome engineering, such as the dominance of the NHEJ repair pathway, the insufficient availability of repair templates due to delivery of limiting amounts and susceptibility to degradation by cellular nucleases, and the lack of control over the spatial and temporal availability of the repair-template at the site of the DSBs[16–19].

Several strategies have been used to enhance HDR-mediated genome engineering in plants including: inhibiting the NHEJ pathway, thereby indirectly enhancing the frequency of HDR; expressing or activating the HDR machinery; stabilizing the repair templates; and regulating and limiting the DNA nuclease activities to specific cell types or cell cycle phases where HDR is naturally active[18,20]. Alternatively, to deliver a sufficient amount of repair-template to the plant cell nucleus, viral replicons and physical methods such as polyethylene glycol treatment of plant protoplasts or particle bombardment of the repair-template were applied to enhance HDR in plants[16,21,22].

To enhance HDR, we previously developed an RNA-templated CRISPR/Cas9 system[15]. A similar approach using an RNA transcript for DNA repair in conjunction with CRISPR/Cas was also applied by Li et al.[23]. These approaches increased the efficiency of HDR in plants, but they suffered from inconsistencies, limitation of the cargo capacity of the repair-template, lack of broader applicability in different crop species, and the uncertain fate of the genetic elements used in such modification. More recently, new systems have aimed to bring the DNA repair-template directly into the proximity of the DSB, using the HUH endonuclease and SNAP-tag fusion to Cas9, or indirectly via the Cas9-biotin-avidin system; these systems enhanced the efficiency of HDR in mammalian cells[20,24–26]. However, these methods are expensive, and not well-explored in plants.

In plants, where the HDR machinery is not active throughout the cell cycle, increasing the proximity of the repair templates to the DSBs may improve the efficiency of HDR. However, so far, no efficient and scalable HDR method has been developed in plants. The strategy of covalently tethering the DNA repair-template to the Cas9 endonuclease could enhance the efficiency of HDR in plants as it can substantially increase the local concentration of repair DNA template.

Intriguingly, we found a potentially useful tool for enhancing HDR in *Agrobacterium tumefaciens*. Agrobacteria produce various Vir proteins to transfer the T-DNA complex into the plant cell[27–29]. Upon induction, the relaxosome complex of VirD1 and VirD2 attaches to the Ti plasmid, and VirD2 cleaves the bottom strands of the Ti plasmid in the left and right border (LB and RB, respectively). Moreover, VirD2 remains covalently attached through tyrosine 29 to the 5′ end of the single-stranded T-DNA (ssT-DNA)[30,31]. In the plant cell, the nuclear localization signal (NLS) in VirD2 leads the ssT-DNA into the nucleus and helps integrate the T-DNA in the plant genome[32,33]. VirD1 and VirD2 overexpressed in *Escherichia coli* or transiently expressed in plant cells were able to process T-DNA, and VirD2 remained covalently bound to the 5′ end of the processed ssDNA[34,35]. Furthermore, VirD2 can nick and covalently bind to the RB-containing ssDNA without the assistance of VirD1[36,37]. Therefore, we hypothesized that, if combined with the nuclease activity of Cas9, VirD2 could bind to a wide range of repair templates of various lengths, to enhance the rate of HDR in plants by bringing the repair-template in proximity to the DSB. The relaxase activity of VirD2 could also help with the proper integration of the repair templates at the target site.

We preferred VirD2 over three other strategies used in mammalian cultured cells[24–26] because VirD2 requires only a 25-nt sequence for binding, and after covalently binding to the RB sequence, VirD2 only adds three extra nucleotides to the end of the repair-template[38]. Additionally, VirD2 naturally evolved with plants to engineer plant genomes and VirD2 bound T-DNAs with homologous ends or VirD2-meganuclease fusions bound to T-DNAs can enhance gene targeting via HDR[39].

Here, we developed a chimeric Cas9-VirD2 protein that combines the functions of Cas9 and VirD2. We demonstrate the use of Cas9-VirD2 to covalently bind to the repair-template via VirD2, and to generate DSBs via Cas9, to enhance the rate of HDR in planta. Using this system, we precisely modified the wild-type rice (*Oryza sativa*) *ACETOLACTATE SYNTHASE* (*OsALS*) allele to one that confers herbicide resistance and successfully recovered herbicide-resistant plants. Similarly, using the VirD2.Cas9 module we precisely edited the *CAROTENOID CLEAVAGE DIOXYGENASE 7* (*OsCCD7*) locus in rice to modify plant architecture. Also, we successfully used this system to produce a precise in-frame insertion of the HA epitope sequence at the C terminus of *HISTONE DEACETYLASE* (*OsHDT*). Our Cas9-VirD2 system has the potential to expand precision genome engineering in plants and across diverse eukaryotic species.

## Results

**Cas9-VirD2 fusions efficiently bind to DNA templates and cleave DNA targets.** We hypothesized that bringing the repair-template in proximity to the DSB would enhance the rate of HDR events in plants (Fig. 1). The Agrobacterium VirD2 relaxase covalently binds, translocates, and helps in the integration of ssDNAs (T-DNA) of diverse length into the nuclear genome[32,40]. Here, we combined the VirD2 binding and integration activities with *S. pyogenes* Cas9 to enhance the rate of HDR in plants. VirD2 was fused in-frame to the N or C terminus of Cas9 in bacterial expression system plasmids. VirD2-Cas9 and Cas9-VirD2 were purified from *E. coli* (Fig. 2a and Supplementary Fig. 1), and sgRNAs were synthesized in vitro (Supplementary Fig. 2).

To confirm that fusion to Cas9 does not affect VirD2 nicking and covalent binding to ssDNA containing the 25-nt RB and LB, both purified fusion proteins were incubated with a >10 kb binary plasmid containing the T-DNA flanked by LB and RB sequences in $Mg^{2+}$ buffer. Cas9 without any fusion was used as a control.

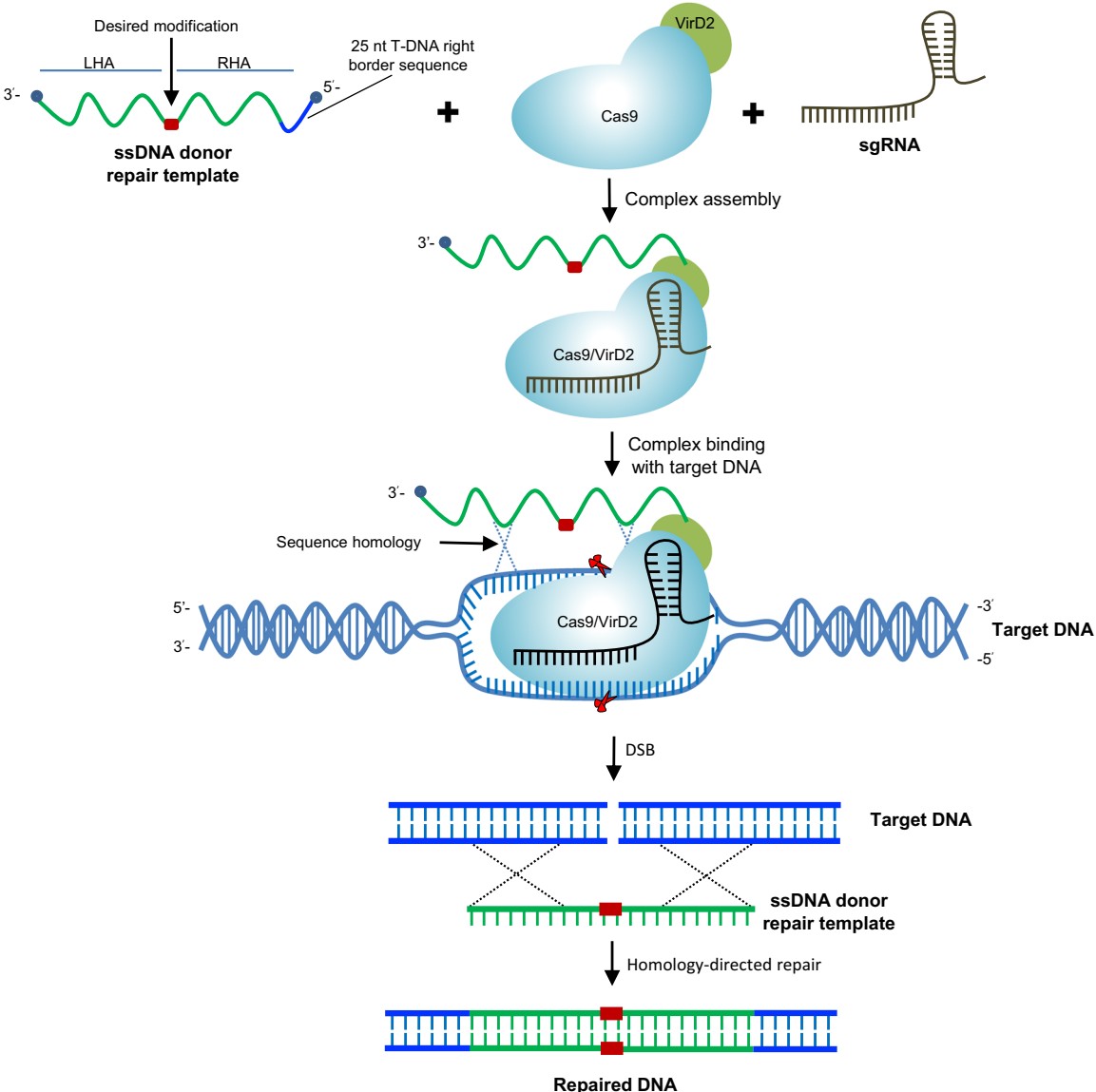

**Fig. 1 Chimeric fusion of Cas9 and VirD2 can facilitate precise genome engineering.** Experimental scheme showing that the Cas9 endonuclease and VirD2 DNA binding activities can be exploited for precise genome engineering of plants. The VirD2 protein in the chimeric Cas9-VirD2 fusion will process and covalently bind the RB-edged ssDNA repair templates. The availability of the repair templates in the proximity of the DNA DSBs created by the sgRNA-guided Cas9 endonuclease may enhance the rate of precise repair via the targeted HDR pathway. Alternatively, the NHEJ pathway will imprecisely repair the DNA DSBs.

Conversion of the supercoiled binary plasmid to completely relaxed and gel-retarded DNAs with increasing concentrations of fusion proteins confirmed the nicking and binding activity of VirD2 on the binary plasmid (Fig. 2b, and Supplementary Fig. 3).

Next, we confirmed the covalent binding of VirD2-Cas9 and Cas9-VirD2 to the 60-nt repair-template containing the 5′-RB (T-RB-60b) by a gel retardation assay. The gel retardation of the protein–DNA complex, even after boiling and running on denaturing SDS-PAGE, confirmed the covalent binding of VirD2-Cas9 and Cas9-VirD2 to the repair-template containing the 5′-RB sequence. By contrast, VirD2-Cas9 and Cas9-VirD2 produced no gel retardation when incubated with a repair-template lacking the RB sequence (T-NRB-60b) (Fig. 2c and Supplementary Fig. 4).

VirD2 can re-ligate ssDNA nicked products in vitro and the reaction appears to saturate at ~40%[37]. Such re-ligations are compensated for by integration of the T-DNA into the plant

genome. Phosphorothioate end modification, with exonuclease to remove the unprotected nicked substrate from the reaction, can enhance the rate of irreversible conjugation of VirD2 with the RB-containing ssDNA[36]. The phosphorothioate end-modified repair templates with RB (mT-RB) were incubated with VirD2-Cas9 in the presence of Exonuclease 1. The removal of the nicked unbound 3′-opened ssDNA from the reaction by the 3′ - 5′ nuclease activity of Exonuclease 1 leads to the complete binding of the repair templates by VirD2-Cas9 (Fig. 2d and Supplementary Fig. 5).

Cas9 nuclease activity remained unchanged by binding different tags and fusion of different domains to the N or C termini[41,42]. To confirm that VirD2 fusion to Cas9 does not compromise Cas9 nuclease activity, we incubated an sgRNA targeting a 20-nt internal site in the target dsDNA substrate with VirD2-Cas9, Cas9-VirD2, or Cas9 in $Mg^{2+}$ buffer. Similar to Cas9, both VirD2-Cas9 and Cas9-VirD2 efficiently cleaved the

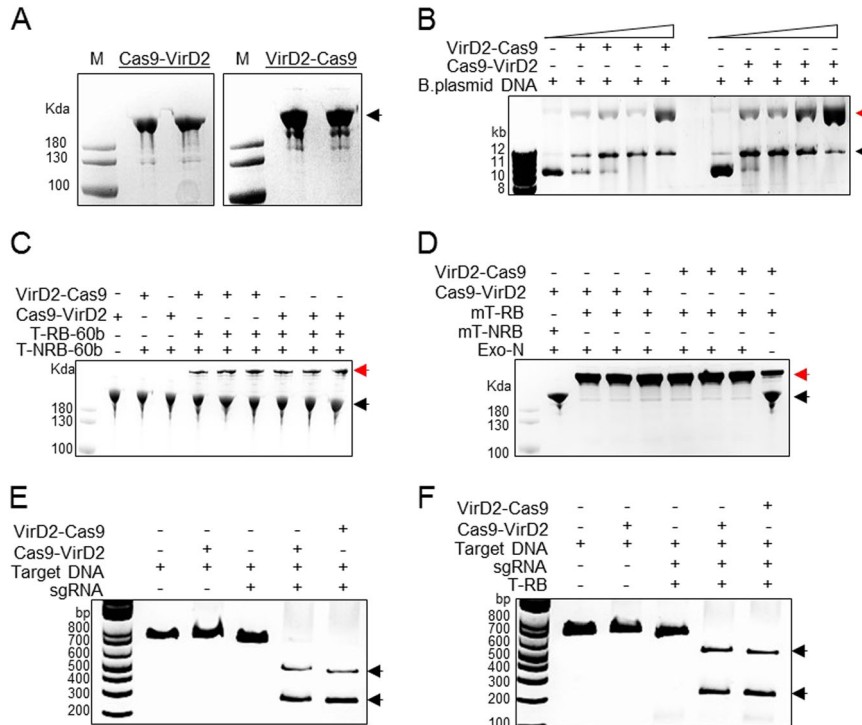

**Fig. 2 Chimeric Cas9-VirD2 fusions efficiently bind to repair template DNA and cleave the DNA target. a** Expression and purification of Cas9-VirD2 and VirD2-Cas9 from BL21(DE3) cells. The HIS column-purified Cas9-VirD2 and VirD2-Cas9 fusion proteins were separated on SDS-PAGE. Both Cas9-VirD2 and VirD2-Cas9 with the exact size of 216 kDa were purified and 1 μg was loaded into the gel for separation. **b** Confirmation of the nicking and relaxase activity of the Cas9-VirD2 and VirD2-Cas9 fusions. The T-DNA vector with the RB and LB was incubated with increasing concentrations of Cas9-VirD2 and VirD2-Cas9. The complex was separated on a 1% TBE agarose gel. The red arrowhead indicates the conversion of the supercoiled plasmid to a completely relaxed gel-retarded DNA structure. **c** Confirmation of the covalent binding of Cas9-VirD2 and VirD2-Cas9 fusions to the repair templates. ssDNA (60 nt) RB sequence (T-RB-60b) and without RB (T-NRB-60b) were incubated with Cas9-VirD2 and VirD2-Cas9 in $Mg^{2+}$ buffer. After incubation, the sample mixture was boiled and separated on denaturing SDS-PAGE. The red arrowhead indicates binding of only the RB-containing repair templates (T-RB-60b) to Cas9-VirD2 and VirD2-Cas9. (**d**) Optimization of the covalent binding of VirD2-Cas9 fusions to the repair templates. ssDNA (60 nt) with RB sequence (T-RB) and without RB sequence (T-NRB) were incubated with VirD2-Cas9 in $Mg^{2+}$ buffer. After incubation for 5 min Exonuclease 1 was added to the sample mixture and incubated for another 25 min, the samples were boiled and separated on denaturing SDS-PAGE. The red arrowhead indicates binding of only the RB-containing repair templates (T-RB) to VirD2-Cas9. (**e** and **f**) Confirmation of the targeted endonuclease activity of Cas9-VirD2 and VirD2-Cas9. The purified Cas9-VirD2 or VirD2-Cas9 proteins and sgRNA with and without repair template were incubated with the target DNA. The reaction mixture was separated on a 2% agarose gel. Arrowheads indicate the proper digestion of the target by the Cas9-VirD2 and VirD2-Cas9-sgRNA complex in the presence and absence of the repair templates.

DNA substrate (Fig. 2e and Supplementary Fig. 6). Next, we tested whether binding of the repair-template to Cas9-VirD2 and VirD2-Cas9 affected the catalytic activity of Cas9. Our results show that following the covalent binding of the repair DNA template, VirD2-Cas9 and Cas9-VirD2 retain a high nuclease activity and would be sufficient for creation of a DSB at the target locus (Fig. 2f and Supplementary Fig. 6).

**Cas9-VirD2 fusions mediate efficient editing of the *ALS* gene in rice calli.** To test the activities of Cas9-VirD2 and VirD2-Cas9 in plant cells, we cloned the plant codon-optimized Cas9 and VirD2 fusions along with a construct for expressing the sgRNAs into vectors for expression in rice protoplasts (Supplementary Fig. 7). Plasmids expressing the fusion proteins along with the Cy5-labeled repair templates (T-RB and T-NRB) were transfected into rice protoplasts. Using anti-Flag antibody-bound agarose beads, the Cas9-VirD2-T-RB complexes were pulled down. The experimental results demonstrate the efficient binding of the RB containing repair templates (Cy5-T-RB) in plant cells expressing VirD2-Cas9 compared to the repair templates having no RB (Cy5-T-NRB), which were not pulled down (Fig. 3a and Supplementary Fig. 8).

Next the plant codon-optimized Cas9-VirD2 fusions along with sgRNAs were cloned into plant expression vectors (Fig. 3b). For in vivo activity the plasmids expressing either Cas9-VirD2 or VirD2-Cas9, or Cas9 only, and sgRNAs targeting *ALS* were bombarded into regenerating rice calli. From each set of bombarded tissue, 24 calli were pooled into one sample and DNA was extracted for molecular analysis. PCR amplicons flanking the target sequence were subjected to the T7 endonuclease (T7EI) assay, which detects mismatches in a sequence, to test for successful editing. The patterns of T7EI digestion of the PCR amplicons encompassing the target sequence confirmed the in vivo nuclease activity of Cas9-VirD2 and VirD2-Cas9 (53 and 47%, respectively) compared to the 56% nuclease activity of Cas9 alone (Fig. 3c and Supplementary Fig. 9).

To validate the results of the T7EI digestion assay showing successful editing by Cas9-VirD2 and VirD2-Cas9 at the target region, the PCR amplicons were cloned into pJet2.1, and the extracted plasmids were subjected to Sanger sequencing. Sanger sequencing reads with indels at the sgRNA targeting site confirmed that the nuclease activity of Cas9-VirD2 and VirD2-Cas9 was similar to that of Cas9 (Fig. 3d and Supplementary Fig. 10).

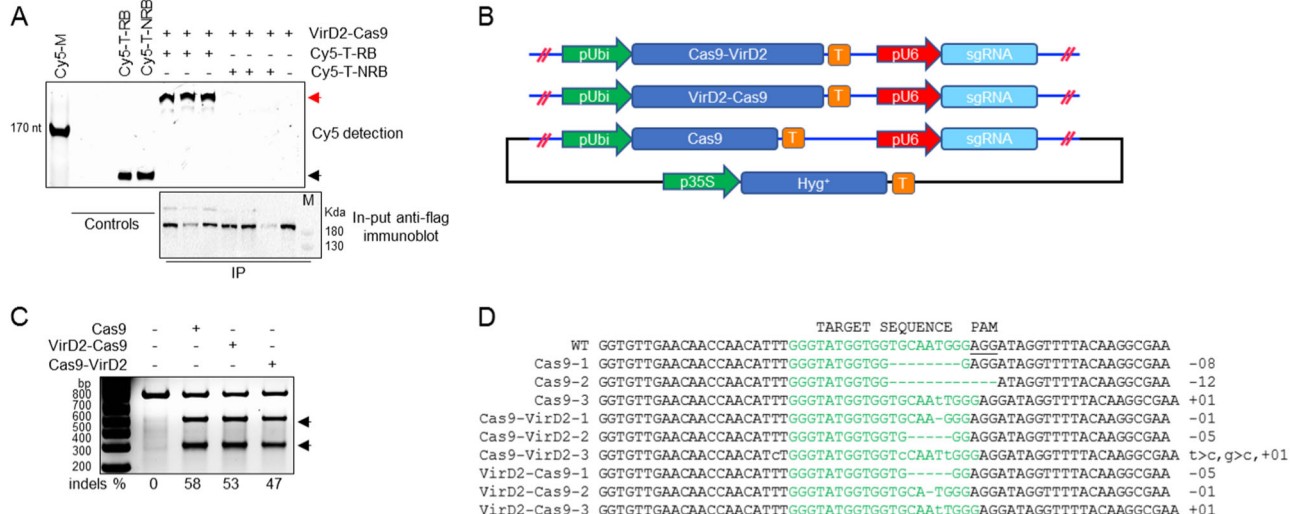

**Fig. 3 Chimeric Cas9-VirD2 fusions mediate efficient editing of *ALS* in rice calli. a** Co-immunoprecipitation of the repair template with VirD2-Cas9 from rice protoplasts. Rice protoplasts transfected with Cy5-labeled repair templates (T-RB, templates with RB and T-NRB, templates without RB) and plasmid expressing VirD2-Cas9 using PEG-mediated delivery. VirD2-Cas9 was immunoprecipitated with anti-Flag bound agarose beads. Part of each sample was run on a 6% agarose gel and imaged (upper panel) for Cy5. The unlabeled (T-RB) and Cy5-labeled (Cy5-T-RB) repair templates were used as controls. A part of the sample was run on SDS-PAGE and the immunoblot was developed for the Flag tag (VirD2-Cas9-Flag) as input with anti-Flag and anti-mouse secondary antibody. **b** Schematic diagram of the in planta expression plasmids. The plant codon-optimized Cas9-VirD2 and VirD2-Cas9 complexes were cloned into a plant expression vector under the control of the *Ubiquitin* promoter followed by a terminator, and sgRNAs were cloned under the *U6* promoter followed by a termination signal. **c** Confirmation of the in vivo endonuclease activity of the Cas9-VirD2 and VirD2-Cas9 fusions. The T7EI assay was conducted with 200 ng of target-flanking purified PCR product. A high rate of Indels was detected by the T7EI digestion assay of the samples expressing Cas9 (58%), Cas9-VirD2 (53%), or VirD2-Cas9 (47%) and sgRNA. T7EI-treated samples were separated on a 2% agarose gel. Arrowheads indicate the corresponding T7EI digestion at the Indel created by the endonuclease activity of the Cas9 fusions. **d** Alignments of the Sanger sequencing reads of the target-flanked PCR product cloned into the pJet2.1 plasmid. The top line shows the wild-type sequence with the target sequence in green and the PAM sequence underlined. Alignment analysis shows the presence of Indels at the target sites. The induced Indels are represented by numbers to the right of each lane.

**Cas9-VirD2 fusions with phosphorothioate-modified templates mediate efficient HDR at the target locus.** Colocalization of the repair-template to the DSB increased the successful editing rate by 24-fold in human cell lines[25]. To test our hypothesis that the proximity of the repair-template to the DSB can enhance HDR in plants, we tethered the repair-template with Cas9 via VirD2 (Fig. 1). As an initial target, we selected the *ALS* locus in the nuclear genome of rice. ALS is involved in branched-chain amino acid biosynthesis, and only targeted conversions of W548L and S627I renders rice resistant to the herbicide bispyribac[43]. To make a DSB at the targeted locus and use the cellular machinery for efficient HDR, we designed sgRNA and different repair templates including the desired sequence for making the W548L edit, flanked by short right and left homology arms (50 bp) with or without the RB for binding to VirD2-Cas9 and Cas9-VirD2 (Supplementary Fig. 11). The repair templates were designed to include the W548L edit for herbicide resistance, modified PAM sequence to avoid retargeting, and to produce a MfeI recognition sequence upon successful editing, a three-nucleotide difference from the wild-type allele for allele-specific PCR and a single-nucleotide divergence for the confirmation of each repair-template (Fig. 4a).

In our experimental design, we bombarded the repair templates and plasmids expressing Cas9, VirD2-Cas9, or Cas9-VirD2, along with the hygromycin resistance gene, and sgRNA from promoters compatible with plant transcription. To protect the repair templates from cellular nucleases and to ensure the stability and availability of the repair templates to be used by the cellular-expressed Cas9-VirD2 fusions, we included the phosphorothioate linkage at the 5′ and 3′ ends in our repair templates (Supplementary Fig. 11). Chemical modifications of the DNA

repair-template for HDR, like 5′-phosphorylation and incorporation of the phosphorothioate linkage at the 5′ and 3′ ends, stabilize repair templates in mammalian cultured cells[44]. The repair templates (Supplementary Fig. 11) with and without end phosphorothioate linkages and 5′-RB sequence (T-RB, T-NRB, mT-RB, and mT-NRB) were custom synthesized.

To confirm that the phosphorothioate linkage end modification can protect the repair templates from cellular nucleases and the repair templates would be available for HDR at the Cas9-sgRNA targeted site, we introduced the repair templates and the respective plasmids into the rice calli by bombardment. Following bombardment and recovery, the rice calli were selected on hygromycin for 4 weeks and 24 random calli were selected for molecular analysis from each set. Total DNA was isolated individually from 24 hygromycin-selected rice calli for molecular analysis. Using allele-specific PCR, our results confirmed that, compared to the unmodified repair templates, the phosphorothioate end-modified repair templates were more efficient, as a high number of calli showed successful editing at the target site compared to no editing with repair templates that lack the end modification (Fig. 4b and supplementary Fig. 12).

Next, to confirm that bringing these modified repair templates to the DSB could enhance the rate of successful editing, the rice calli were bombarded with plasmid DNA expressing Cas9-VirD2 and an sgRNA targeting *ALS*, along with phosphorothioate end-modified repair templates with and without the RB sequence. Allele-specific PCR from DNA isolated from 24 individual bombarded rice calli for each condition showed that 20.8% of the calli had cells that had undergone successful editing in the samples bombarded with Cas9-VirD2 and the end-modified RB-containing repair templates (mT-RB) (Fig. 4b, Table 1 and

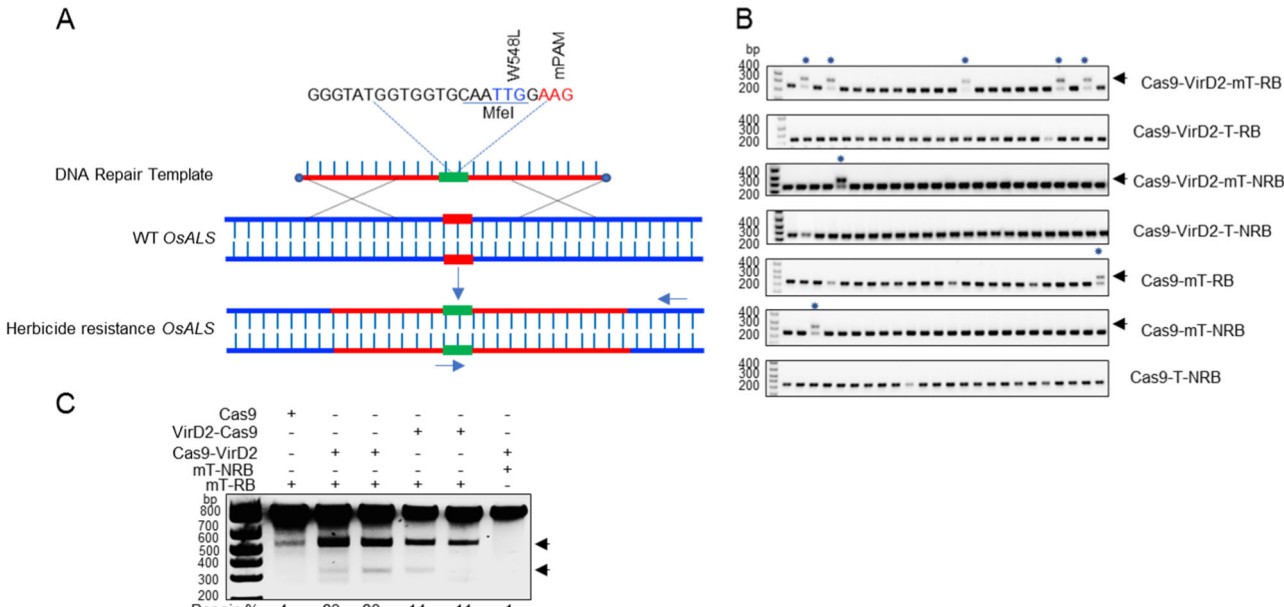

**Fig. 4 Chimeric Cas9-VirD2 fusions with phosphorothioate-modified repair templates make efficient edits at the *ALS* target locus. a** Schematic of the desired edit (green) in the wild-type *OsALS* allele. The desired modification (W548L), the MfeI recognition site, and the modified PAM sequence in the repair template are shown. The repair template (red line, right and left homology arms) with the desired modification (green) was used to make edits at the target site (black line). The forward allele-specific PCR primer is represented by an arrow corresponding to the desired modification and the reverse primer outside the homology arm sequence. **b** Allele-specific confirmation of successful editing in the rice callus. The allele-specific PCR from the DNA extracted from 24 individual calli bombarded with phosphorothioate-modified repair templates and unmodified repair templates with and without RB sequence (T-NRB, mT-NRB, T-RB and mT-RB) with Cas9 or Cas9-VirD. The phosphorothioate-modified repair templates are more stable than the unmodified repair templates and enhance the rate of HDR in combination with Cas9-VirD2. The amplification of the exact size 417 bp fragment only in the VirD2-Cas9 or Cas9-VirD2 with mT-RB samples by allele specific PCR (indicated by the arrow head) confirmed the presence of the herbicide resistance allele. The lower band is a nonspecific PCR amplification and present in all samples. **c** Molecular confirmation of the exact insertion of the desired nucleotide sequence at the target locus via MfeI digestion. Restriction digestion of the target-flanked PCR product from the DNA of pooled selected calli ($N = 100$). Using ImageJ software, the rate (in %) of the exact repair (MfeI recognition site insertion) was calculated and is represented below the corresponding lane. Cas9 alone was used as the baseline control. The arrowheads represent the MfeI-digested fragments.

| Table 1 Representing the enhanced rate of successful editing with the phosphorothioate-modified repair templates (mT-RB) and Cas9-VirD2. | | | |
| --- | --- | --- | --- |
| **Clones and repair templates** | **Repaired + ive calli** | **Total no of calli** | **Calli with modified allele %** |
| Cas9-VirD2 + mT-RB | 5 | 24 | 20.8% |
| Cas9-VirD2 + T-RB | 0 | 24 | 0% |
| Cas9-VirD2 + mT-NRB | 1 | 24 | 4.1% |
| Cas9-VirD2 + T-NRB | 0 | 24 | 0% |
| Cas9 + mT-RB | 1 | 24 | 4.1% |
| Cas9 + mT-NRB | 1 | 24 | 4.1% |
| Cas9 + T-NRB | 0 | 24 | 0% |

Supplementary Fig. 13). This represents a five-fold increase compared with Cas9, as only 4% of the calli showed successful editing in samples bombarded with Cas9 and the end-modified repair-template containing the RB (mT-RB), or with Cas9-VirD2 and repair templates without the RB (mT-NRB) (Fig. 4b and Supplementary Fig. 13).

To validate our data for Cas9-VirD2 and VirD2-Cas9, 100 calli that were positive for successful editing by allele-specific PCR were selected from each set of conditions and grown for 4 more weeks on hygromycin. Then, the calli were pooled and DNA was extracted for allele-specific PCR. The allele-specific PCR amplified fragments of the expected size from the calli expressing either Cas9-VirD2 or VirD2-Cas9 and sgRNA; surprisingly, the Cas9 samples produced similar fragments (Supplementary Fig. 14). Therefore, we evaluated the rate of successful editing via MfeI digestion assays. Based on our design, if used

successfully in HDR, our repair-template will create an MfeI restriction enzyme recognition site (Fig. 4a). Restriction digestion of the PCR fragment flanking the repair site with MfeI demonstrated a clear enhancement of 5.7-fold (23% in Cas9-VirD2) and 3.5-fold (14% in VirD2-Cas9) of the insertion of the MfeI recognition site indicative of successful editing, compared to 4% with Cas9 only (Fig. 4c and Supplementary Fig. 15).

Furthermore, we conducted deep amplicon sequencing to validate our allele-specific PCR results and determine the approximate number of calli needed for regeneration and recovery of a reasonable number of plants harboring the edited allele. Subsequently, we compared the frequency of the presence of the repaired allele using Cas9, Cas9-VirD2, and VirD2-Cas9. Our data showed that both Cas9-VirD2 and VirD2-Cas9 enhanced the editing frequency by 15-fold in calli bombarded with the modified repair templates containing the RB sequence

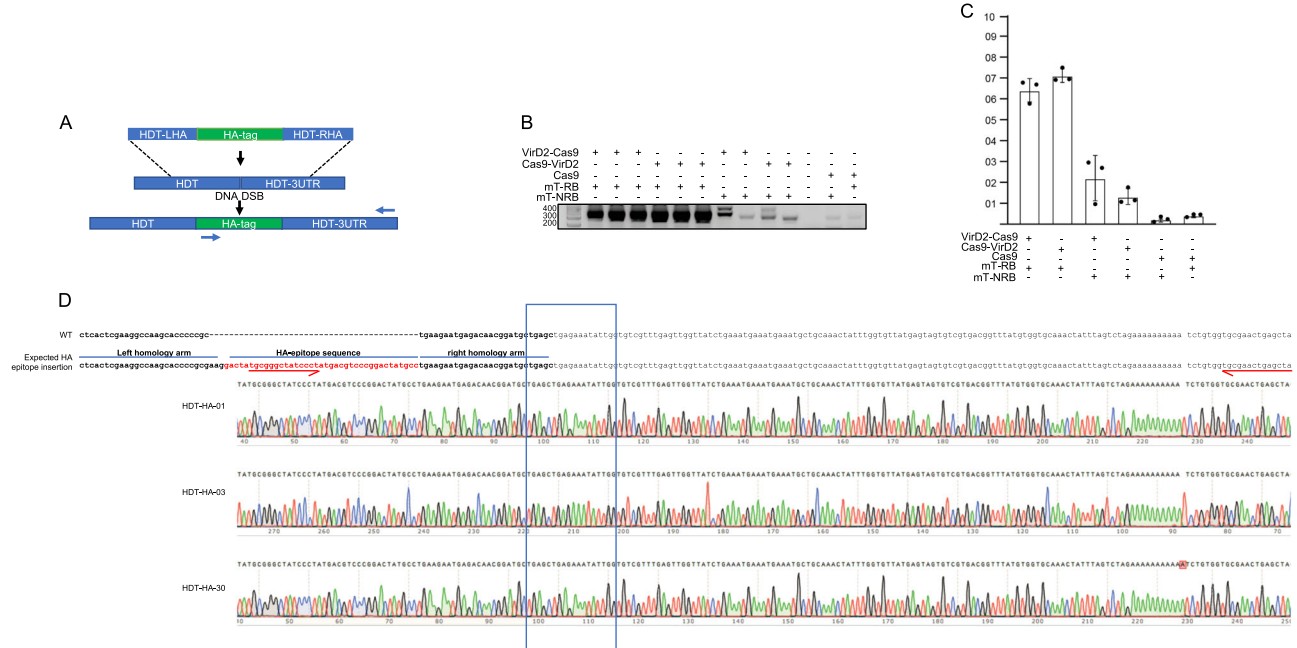

**Fig. 5 The Cas9-VirD2-coupled HDR system efficiently inserted the HA epitope into the C terminus of *OsHDT*. a** Schematic of the in-frame insertion of the HA epitope at the C terminus of *OsHDT*. The short right and left homology arms (blue) flanked the HA epitope (green). The DNA DSB is shown by a line, and the site of the insertion is indicated by a black arrow. The dotted lines represent the targeted HDR site. The forward allele-specific PCR primer corresponding to the HA epitope DNA sequence and the reverse primer outside the homology arm sequence are represented by red arrows. **b** Confirmation of the insertion of the HA epitope at the C-terminal end of *OsHDT*. The allele-specific PCR (22 cycles) from the DNA extracted from pooled (N = 100) calli bombarded with the end modified repair template (left homology arm–HA epitope–right homology arm) with and without the RB (mT-RB and mT-NRB) and Cas9-VirD2, VirD2-Cas9, or Cas9. **c** Representation of the enhanced rate of insertion of the HA epitope using the repair templates containing the RB sequence and Cas9-VirD2 or VirD2-Cas9. **d** Confirmation of the targeted insertion of the HA epitope at the C terminus of *OsHDT* via Sanger sequencing. The HA epitope-specific forward primer PCR product from the DNA of the pooled calli (N = 24) was cloned into the pJet2.1 plasmid and subjected to Sanger sequencing. The exact alignment of the Sanger sequence reads confirmed the successful editing at the C-terminal end of *OsHDT* in the calli bombarded with the repair templates with the RB and VirD2-Cas9. Box in Sanger sequencing chromatogram represent the precise repair and accurate end junction of the repair template insertion at c-terminus of *OsHDT*.

(mT-RB) and Cas9-VirD2 and VirD2-Cas9 compared to repair templates containing no RB (mT-NRB) with Cas9-VirD2 (up to 2.8-fold) and Cas9 only with repair templates containing the RB sequence (mT-RB) (up to 2-fold) (Supplementary Fig. 16).

**The Cas9-VirD2-coupled HDR system efficiently inserted the HA epitope into the C terminus of *OsHDT*.** To broaden the applicability of our module of coupling the repair templates to the DSB for precise and versatile genome engineering, we investigated whether we can achieve precise domain knock-in and tag the HA epitope in-frame into *OsHDT*. Therefore, we designed a sgRNA targeting the C-terminal end of the *HDT* sequence and a corresponding repair-template that contained the HA epitope flanked with short homology arms of the *HDT* gene sequence and 3′-untranslated region (UTR) with (mT-RB) and without the RB sequence (mT-NRB) (Fig. 5a). The precise repair would produce HDT fused to HA in both the DNA and polypeptide sequence. We delivered the plasmids expressing Cas9-VirD2, VirD2-Cas9, or Cas9 and the *HDT*-targeting sgRNA with the repair templates containing the HA epitope. Samples were collected in pools (N = 100 calli/pool) from the calli transformed with the Cas9-VirD2-, VirD2-Cas9, or Cas9-expressing vectors after 21 days on hygromycin selection media.

For confirmation of successful editing at the C terminus of *HDT*, we used allele-specific PCR with a primer set where the forward primer binds specifically to the HA epitope sequence (not present in the rice genome) and the reverse primer binds outside of the repair-template in the 3′-UTR of the *HDT*

sequence (Fig. 5a). We opted for quantitative, low-cycle PCR (22 cycles), and the results indicated an enhanced ratio of successful editing with Cas9-VirD2 and VirD2-Cas9 compared to Cas9 alone with the same sgRNA and repair templates (Fig. 5b, c and Supplementary Figs 17, 18). To confirm the successful insertion of the HA epitope at the C terminus of *HDT*, the allele-specific PCR product was cloned and subjected to Sanger sequencing. Sanger sequencing reads clearly confirmed the in-frame tagging of *HDT* with the HA epitope (Fig. 5d and Supplementary Fig. 19).

To determine the exact ratio of successful editing to tag *OsHDT* with the HA epitope, the HA epitope-flanking PCR-amplified fragments were subjected to amplicon sequencing using the TruSeq platform. Consistent with our molecular data, the TruSeq data confirmed an enhanced rate (up to 8.7-fold) of successful editing in samples expressing VirD2-Cas9 compared to VirD2-Cas9 (0.01 fold) and Cas9 (0.2 fold) with repair templates with no RB sequence (Supplementary Fig. 20a, b).

**Engineering efficient and heritable herbicide resistance in plants via chimeric Cas9-VirD2 fusions.** After confirming the rate of HDR with our repair templates coupled with the Cas9-VirD2 system at the cell/tissue levels, we then attempted to regenerate herbicide-resistant rice plants. It would be possible to regenerate herbicide-resistant rice calli on bispyribac-containing media but this may lead to herbicide resistance derived from spontaneous mutation and selection pressure, not through successful editing. To avoid this rare possibility, we regenerated the rice calli on hygromycin-containing media, as the plasmid also

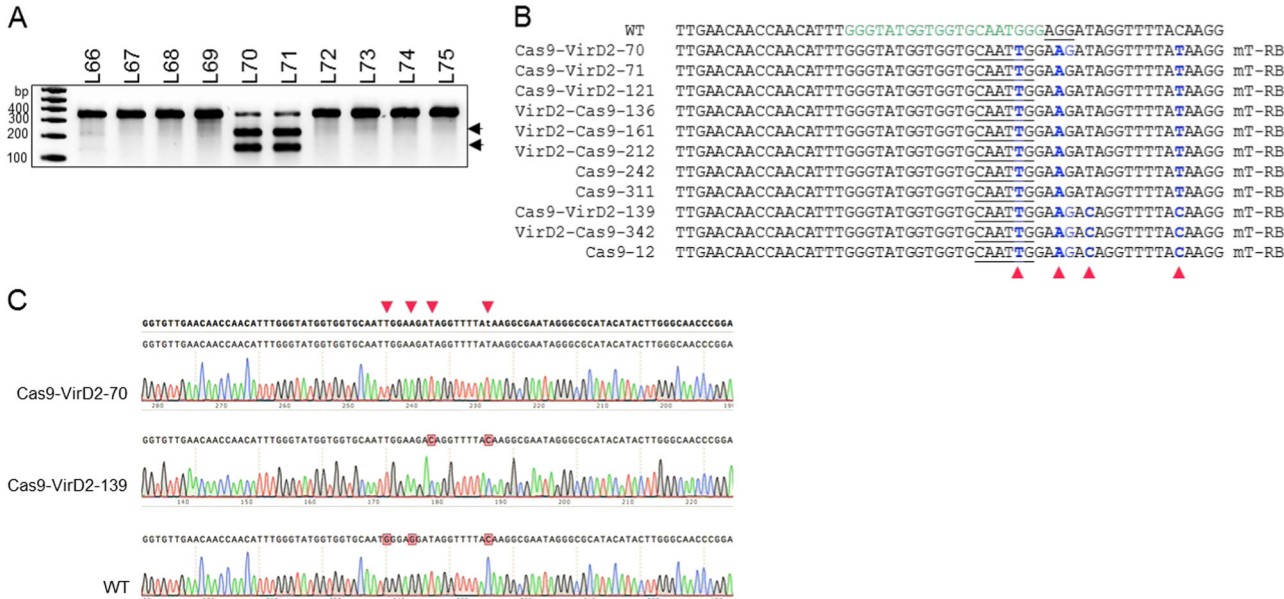

**Fig. 6 Engineering herbicide resistance in rice. a** Confirmation of the presence of the herbicide resistance allele in regenerated rice plants. Genomic DNA was extracted from individual regenerated plants, and the target-flanked PCR-amplified fragment was subjected to MfeI digestion. MfeI digestion confirmed a high rate of successful editing in the plants regenerated from calli bombarded with repair templates containing the RB sequence and Cas9-VirD2 or VirD2-Cas9. **b, c** Alignment of the Sanger sequencing reads and representative chromatograms of the plants that had been confirmed by MfeI digestion. The PCR-amplified fragments from the individual plants were cloned into the pJet2.1 plasmid and subjected to Sanger sequencing. The repair-specific nucleotide modifications are shown in blue, and their exact locations are indicated by arrowheads (red). A representative chromatogram also shows the exact repair at the targeted locus.

**Table 2 Comparison of the successful editing rate between Cas9-VirD2, Cas9-VirD2, and Cas9 with repair templates, with and without the RB. Both Cas9-VirD2 and VirD2-Cas9 with repair templates with the RB demonstrated an efficient rate of HDR at the target site.**

| Clones and repair templates | Fully repaired | Partial repair | Total T0 plants | Repair frequency |
|---|---|---|---|---|
| Cas9-VirD2 + mT-RB | 8 | 3 | 81 | 9.87% |
| VirD2-Cas9 + mT-RB | 3 | 1 | 31 | 9.67% |
| Cas9-VirD2 + mT-NRB | 1 | 0 | 51 | 1.96% |
| VirD2-Cas9 + mT-NRB | 1 | 0 | 63 | 1.58% |
| Cas9 + mT-NRB | 3 | 1 | 186 | 1.61% |
| Cas9 + mT-RB | 2 | 1 | 128 | 1.56% |

The successful editing rate was calculated by the exact number of plants showing the proper conversion of the wild-type *ALS* allele to the herbicide-resistant *ALS* allele in each set of regenerated plants

has the hygromycin resistance gene. ALS is biologically essential, key enzyme of the amino acid biosynthetic pathway, and only targeted sequence modifications leading to amino acid substitutions (W548L and S627I), other than the wild type amino acids, will enable the rice regeneration. Approximately 800 to 1000 calli that had been confirmed by allele-specific PCR were transferred to hygromycin selection media, and 40 to 120 plants were regenerated from each set. DNA was extracted from individual plants, and the PCR amplicons encompassing the repair site were treated with MfeI (Fig. 6a and Supplementary Fig. 21).

Genotyping by MfeI digestion (Fig. 6a) confirmed a recovery of up to 4.6-fold higher (9.87%) and 4.5-fold (9.67%) successfully edited plants with Cas9-VirD2 and VirD2-Cas9, respectively, when used with the phosphorothioate end-modified templates containing the RB sequence (mT-RB). When Cas9-VirD2 and VirD2-Cas9 were used with templates having no RB sequence (mT-NRB), we only recovered 1.96% and 1.58% successfully edited plants, respectively. Similarly, Cas9 used with a template containing the RB sequence (mT-RB) or Cas9 used with a template without the RB (mT-NRB) resulted in the recovery of

only 1.56% and 1.61% successfully edited plants, respectively (Table 2).

Next, the PCR amplicons encompassing the repair site were cloned into pJet2.1 and the extracted plasmids were subjected to Sanger sequencing to confirm the repair at the nucleotide level. Sanger sequencing reads confirmed the successful editing of the wild-type *ALS* allele to the herbicide resistance allele with the W548L modification. Sanger sequence alignment also confirmed all other changes that we included in the repair templates to modify the PAM sequence to avoid retargeting, insertion of the MfeI recognition sequence, three-nucleotide change from the wild-type allele for allele-specific PCR, and a single-nucleotide change for the confirmation of each repair-template (Fig. 6b, c and Supplementary Fig. 22).

To confirm that engineered herbicide resistance via the Cas9-VirD2 coupled template system is heritable, seeds collected from the T₀ regenerated plants were germinated on media containing bispyribac. Germination of seedlings with normal roots, green leaves, and proper segregation confirmed the heritability of the herbicide resistance allele (Fig. 7a and Supplementary Fig. 23). To confirm the heritability of the engineered sequence at the

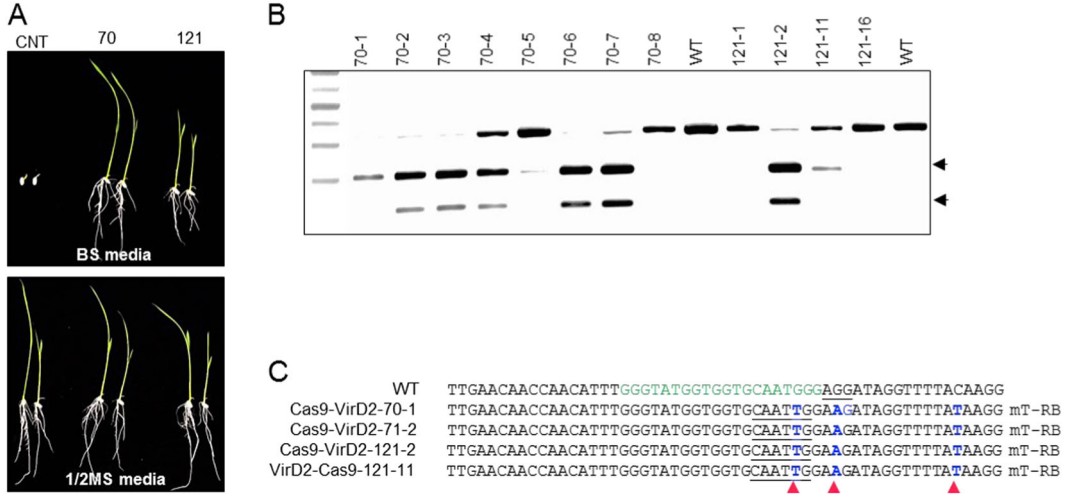

**Fig. 7 Heritability of the engineered herbicide resistance in rice plants. a** Confirmation of the herbicide resistance in the T₁ progenies of rice plants. Seeds collected from line 70 and line 121 were germinated on vertical plates with 1/2 MS media containing bispyribac. Seeds from the plants expressing only Cas9 were used as a control. Plants with the modified *ALS* allele showed proper root and shoot development. Plants expressing only Cas9 with repair templates (CNT) were used as control. **b** Molecular confirmation of the modified *ALS* allele. Genomic DNA was extracted from individual T₁ plants, and the target-flanked PCR-amplified fragment was subjected to MfeI digestion. MfeI digestion confirmed the proper conversion of the *ALS* allele to the herbicide-resistant allele in the selected plants. Arrowheads indicate the MfeI-digested fragment. **c** Alignment of the Sanger sequencing reads of the selected lines 70-1, 70-2, 121-02, and 121-11. The PCR-amplified fragments from the individual plants were cloned into the pJet2.1 plasmid and subjected to Sanger sequencing. The exact locations of the specific nucleotide modifications are indicated by red arrowheads.

molecular level, the DNA was extracted and herbicide resistance allele-specific PCR was conducted. The herbicide resistance allele-specific PCR results confirmed the heritability of the targeted modifications (Supplementary Fig. 24). To confirm in more detail, a DNA fragment flanking the target site was amplified by PCR. Restriction digestion of the resulting PCR product with MfeI produced the appropriately sized digested fragments. These results demonstrated the heritability of the herbicide resistance *ALS* allele (Fig. 7b and Supplementary Fig. 25). Next, we confirmed the heritability of the exact replacement of nucleotides for herbicide resistant *ALS* allele, and precise junctions formation in the selected lines at the nucleotide level by Sanger sequencing of the *ALS* locus These data indicate that our Cas9-VirD2 fusion proteins facilitate and improve HDR frequencies and the edits made are precise and heritable in plants. (Fig. 7c, d and Supplementary Fig. 26).

**Engineering plant architecture via Cas9-VirD2 fusions**. After confirming that Cas9-VirD2 fusions coupled with the RB-containing repair templates can be used for precise gene targeting in plants, we applied this module for engineering rice plant architecture. Engineering of plant architecture can produce a visible trait, allowing easy screening for successful editing events. We select *CAROTENOID CLEAVAGE DIOXYGENASE 7* (*OsCCD7*) in rice for engineering, as OsCCD7 controls a key step in strigolactone biosynthesis[45]. Precise editing (P596L) or loss-of-function of *OsCCD7* results in rice plants with many tillers and a dwarf stature[6,46]; therefore edits that result from NHEJ and HDR will have the same plant phenotype, but can be differentiated by examining the DNA sequence.

We designed sgRNA and phosphorothioate end-modified repair templates with and without the RB sequence (mT-RB and mT-NRB). Multiple single-nucleotide changes were included in the repair-template: the desired P596L change in *OsCCD7*, modification of the sgRNA binding site and destruction of the PAM to avoid retargeting, insertion of an EcoRI recognition sequence, and a one-nucleotide change outside the sgRNA

binding site for easy identification of the successful editing events. (Fig. 8a).

The repair templates (mT-RB and mT-NRB) and plasmids containing the hygromycin resistance gene and expressing Cas9 or Cas9-VirD2 were bombarded into the rice calli and tissues were selected on hygromycin to regenerate plants. From the whole population of each set, we selected only the high-tillered and dwarf rice plants. These plants could have two separate NHEJ repair events, or have one allele resulting from NHEJ and the other allele repaired via HDR, or have both alleles repaired by HDR. To confirmed successful editing, we used primers flanking the target for PCR and digested the product with EcoR1. EcoRI digestion identified 6 plants (8.8%) with the precise insertion out of 69 high-tillered dwarf plants from the Cas9-VirD2 and RB-containing repair-template (mT-RB) samples, compared to only 1 precisely repaired plant (1.5%) out of 64 selected plants from the Cas9-VirD2 and repair-template without RB (mT-NRB) samples (Fig. 8b, Supplementary Figs 27, 28 and Supplementary Table 1). Out of 46 high-tillered dwarf plants in Cas9 and RB-containing repair-template (mT-RB) no plant was confirmed by EcoRI digestion to have a precise template insertion. Also, we did not observe any single plant with complete digestion by EcoRI, suggesting that all these repaired lines are bi-allelic for alleles resulting from HDR and NHEJ repair (Fig. 8b and Supplementary Figs. 27, 28). To verify the genotype of the T₀ regenerated plants that were confirmed by EcoRI digestion, we amplified, cloned, and sequenced the region flanking the target site from three independent T₀ plants. Sanger sequencing confirmed that these plants have one allele imprecisely repaired and the second allele precisely repaired via HDR (Supplementary Fig. 29).

To verify that the alleles resulting from precise repair (Fig. 8c) are heritable, we selected a bi-allelic line 1567 having the highest number of tillers among the selected lines. Seeds from line 1567 were grown and DNA was extracted for molecular analysis. EcoRI digestion of the PCR product covering the target site confirmed the heritability of the engineered sequence (Fig. 8d, Supplementary Fig. 30 and Supplementary Table 2). Next, to confirm that the high-tillered dwarf plants contained all the sequence

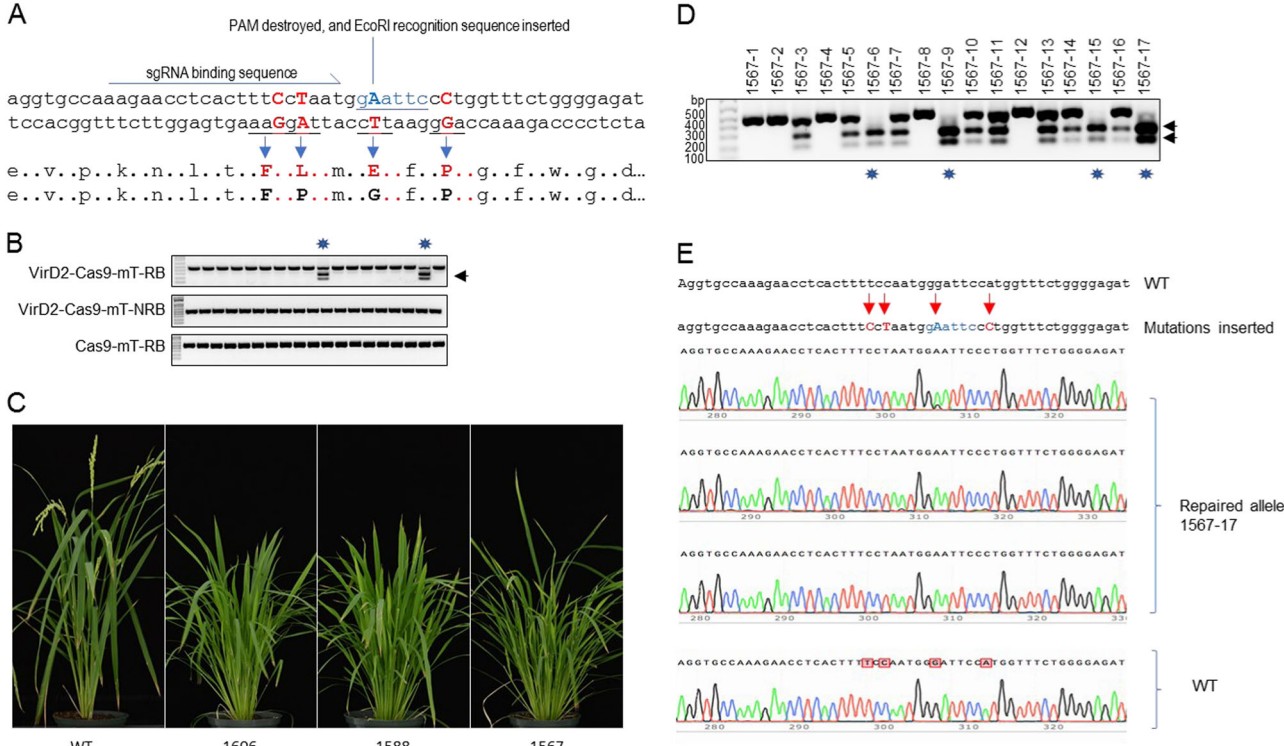

**Fig. 8 Efficient engineering of plant architecture through HDR. a** Schematic of the desired mutations in the wild-type *OsCCD7* allele. Modifications of nucleotides added into the repair templates at the sgRNA binding and PAM sequences to avoid retargeting of the repaired sequence by VirD2-Cas9 are shown. The sgRNA binding sequence is indicated by an arrow and removal of the PAM and insertion of the EcoRI recognition sequence (underlined) is indicated by a line. Nucleotide modifications along with the corresponding amino acids are represented by capital, colored, and bold letters. The respective wild-type amino acids (capital and bold) are also shown. **b** Representative gels of EcoRI digestion for HDR confirmation. Target (*CCD7*) flanking 445 bp PCR amplified fragments were subjected to EcoRI digestion. EcoRI digestion confirmed the proper digestion to produce 270 + 175 bp fragments (indicated by arrowheads) and repair in the plants regenerated from calli bombarded with repair templates with mT-RB and Cas9-VirD2 or VirD2-Cas9. Repair templates without mT-NRB with VirD2-Cas9 and Cas9 with mT-RB repair templates were used as control. Lines confirmed with EcoRI digestion are indicated with asterisks. **c** Photos of plants with engineered modifications. Seed of the progeny plants were grown on soil in the greenhouse and pictures were taken 40 days after germination. **d** Heritability of the repaired modifications. The target (*CCD7*) flanking PCR amplified 445-bp fragments from the progeny plants of the selected lines were subjected to EcoRI digestion. EcoRI digestion produced 270 + 170 bp fragments in lines that had inherited the modification. Four out seventeen plants are homozygous for the EcoRI site. **e** Alignment of the Sanger sequencing reads and representative chromatograms of the EcoRI digestion of confirmed homozygous plants. The repair-specific nucleotide modifications are shown in blue and their exact locations are indicated by arrowheads (red). The representative chromatograms show the exact repair at the targeted locus.

modifications added into the repair-template, we examined four $T_1$ progeny of line 1567 that were confirmed to be homozygous by EcoRI digestion. The PCR products flanking the target site were cloned and sequenced. Alignment of the sequences confirmed the precise insertion of the repair templates along with all four modifications made to the repair templates without any extra nucleotides at the junctions (Fig. 8e and Supplementary Fig. 31).

## Discussion

Developing methods for improving HDR frequencies is of paramount importance for crop bioengineering[47]. Intriguingly, through the use of our Cas9-VirD2 system to bring the repair-template to the proximity of the DSB by VirD2, we achieved a 5- to 6-fold higher repair rate than with Cas9 alone in the precise replacement of the wild-type *ALS* allele with the modified herbicide resistance allele (Fig. 4b, d and Table 1). In our experimental design, we bombarded plasmids expressing either Cas9, VirD2-Cas9, or Cas9-VirD2 and repair templates with a 5′ and 3′ phosphorothioate linkage. The enhanced ratio of repair (4-fold even with Cas9) with 5′ and 3′ phosphorothioate-modified templates demonstrated that these chemical modifications play a major role in the protection of the ssDNA repair templates from

the cellular nucleases, and these repair templates are compatible with the plant cellular HDR machinery. Chemical modifications of the repair DNA for HDR, like 5′-phosphorylation and the incorporation of the phosphorothioate linkage at the 5′ and 3′ ends, were also shown to stabilize repair templates in mammalian cultured cells[44,48]. One interesting phenomenon that would need to be explored in future research is the fate of the chemically modified donor templates, particularly when introduced into rice calli via biolistic transformation. These chemically modified donor templates are more stable and have greater chance of genome-wide random insertion. Such possibilities should be addressed in detail before the VirD2-Cas9 module is used for trait development in crops.

In contrast to delivery of the Cas9-sgRNA RNP complex and the repair-template separately, binding of the repair-template directly or indirectly with the RNP complex was shown to enhance the HDR in animal systems[25,26]. We reasoned that the use of the repair template-tethered RNP would boost the rate of HDR repair and could be a reliable system than other such strategies used for the precise genome engineering of plants (Supplementary Discussion). In plant genetic transformation, VirD2 plays a vital role by binding to the T-DNA, translocating it into the plant cell, and integrating it in the plant nuclear genome.

When combined with Cas9, VirD2 provides an efficient system to bring even a large (almost up to 20 kb) repair-template to the DSB. We preferred VirD2 fusion to Cas9 to enhance HDR over the other three such strategies of bringing the repair templates to the DSB[24–26] or the fusion of the HDR machinery protein like Rad51 to the Cas9[49], because VirD2 could enzymatically bind to a wide range of DNA repair templates and it has the natural ability to integrate ssDNA into the plant genome. This modality was used to recover precisely edited plants via HDR. Our data show that the covalent binding of the repair-template to the bimodular protein (Cas9-VirD2) improved the HDR rate (up to 5.5-fold) in plant cells compared to the use of Cas9 alone. Previously an alternative approach of targeted insertion of the repair-template into the host genome via I-SceI-VirD2 fusion translocation from Agrobacterium was applied to engineer the yeast genome[39]. Similarly, Cas9-VirD2-sgRNA-T-DNA complex translocation would be worth trying in plants. The Cas9-VirD2-sgRNA-T-DNA complex would be much bigger compared to the I-SceI-VirD2.T-DNA used in yeast, but in plants the Agrobacterium translocated Cas9-VirD2-fusion might perform better due to the natural compatibility of VirD2 with plant cells.

VirD2 precisely integrate T-DNA in the genome of yeast and mammalian cells[50–52]; similarly, fusion of VirD2 with the homing endonuclease I-SceI increased the rate of targeted integration of T-DNA in yeast cells[39]. Therefore, we expect that our chimeric Cas9-VirD2 system will be equally adoptable across other eukaryotic species. Moreover, the efficient use of our Cas9-VirD2 system to bring the repair-template into the proximity of the DSB by VirD2, for editing *CCD7* and the recovery of successfully edited multi-tiller dwarf rice plants (Fig. 8) validated the use of this system for precise genome engineering in plants. Our Cas9-VirD2 repair complex can be efficiently applied in gene targeting, gene stacking, or genome engineering of agronomic traits and crop improvement.

## Methods

**Plasmid construction**. The VirD2 open reading frame was PCR amplified from the total DNA isolated from *Agrobacterium tumefacien* strain GV3101 and cloned into the pENTER-D-Topo vector, and the sequence was confirmed by Sanger sequencing. To express Cas9-VirD2 or VirD2-Cas9 under the T7 promoter, the VirD2 open reading frame was fused in-frame to the N or C terminus of Cas9 in pET28a (Addgene plasmid #53261) following the Gibson assembly protocol. For in planta expression, the plant codon-optimized sequence of VirD2 was custom synthesized and cloned in-frame to either the N or C terminus of Cas9 in pRGE32 and pRGEB32 (Addgene plasmid #63142) following the Gibson assembly protocol. sgRNAs as a primer dimer were first cloned into the entry clone under the U3 promoter by Golden Gate cloning with BsaI. The whole cassette, U3::sgRNA, was PCR amplified and cloned into the HindIII and SbfI sites in the pRGEB-Cas9-VirD2 and pRGEB-VirD2-Cas9 vectors. The sequences of all primers used in PCR amplification are provided in Supplementary Table 3.

**Protein purification**. *E. coli* strain BL21 (DE3) (Novagen) was used for the expression of all recombinant proteins Briefly, Cas9, Cas9-VirD2 or VirD2-Cas9 in pET28a plasmid were transformed into BL21 (DE3) cells. Cell were grown in 2L of 2YT media supplemented with the kanamycin at 37 °C to an OD$_{600}$ of 0.7, followed by induction with 0.3 mM isopropyl β-D-thiogalactopyranoside (IPTG) for 12 h at 22 °C. The cells were harvested by centrifugation at 5500$g$ for 10 min, and the resulting cell pellets were re-suspended in 3 ml per 1 g of wet cells in lysis buffer (50 mM Tris-HCl pH (8), 500 mM NaCl, 20 mM Imidazole, 0.1% NP-40, 1 mM PMSF, 5% Glycerol and EDTA free protease inhibitor cocktail tablet/50 ml (Roche, UK)). Cells were lysed enzymatically by adding 2 mg/ml lysozyme and mechanically by sonication. The lysate was clarified by centrifugation at 95,834$g$ for 50 min. The clarified supernatant was then loaded directly onto HisTrap HP 5 ml affinity column (GE Healthcare) that was pre-equilibrated with buffer A (50 mM Tris (pH 8), 500 mM NaCl, 20 mM Imidazole). The column was then washed with 75 ml of buffer A followed by washing with buffer A containing 40 mM imidazole. Bound proteins were eluted with a 75 ml gradient using buffer B (50 mM Tris (pH 7.5), 500 mM NaCl, 500 mM Imidazole) starting from 40 mM to 500 mM imidazole. The peak fractions were pooled and concentrated to 1 ml and manually injected onto HiLoad Superdex 16/600 200 pg gel filtration column (GE Healthcare) that was pre-equilibrated with gel filtration buffer (20 mM Tris (pH 7.5), 200 mM KCl

and 10 mM MgCl$_2$). The eluted fractions were collected, concentrated, flash frozen and stored at −80 °C.

**In vitro digestion of target DNA**. DNA fragments of the target genes (*OsALS*, *OsACTIN*, and *OsHDT*) were PCR amplified from the genomic DNA of rice (*Oryza sativa*) with their respective sets of primers (Supplementary Table 3). The target DNA (300 ng) was incubated at 37 °C with 1 μg of sgRNA and 1 μg of Cas9-VirD2 or VirD2-Cas9 for 1 h. Cas9 alone was used as a control. DNA fragments were separated on 2% agarose gels.

**Repair templates**. The repair templates, phosphorothioate-modified and unmodified (ssDNA 100 bp with and without RB sequence were custom synthesized (IDT). Sequences of the repair templates with and without the RB are provided in Supplementary Table 3.

**In vitro binding of the repair templates to the Cas9-VirD2 and VirD2-Cas9 fusion proteins**. For covalent binding of the Cas9-VirD2 and VirD2-Cas9 fusion proteins to the repair templates, the repair templates (1 μM of ssDNA or 300 ng of dsDNA) were incubated with 1 μg of either Cas9, Cas9-VirD, or VirD2-Cas9 at 37 °C for 30 min. For complete binding 1 μM of ssDNA or were incubated with 1 μg of Cas9-VirD at 37 °C in MgCl2 buffer for 5 min and 1 ul of Exonuclease 1 (*E. coli*, NEB) was added following manufacturer's protocol and the reaction was incubated for 25 more minutes. The DNA-protein complex was separated on 6% SDS-PAGE for the ssDNA-protein complex or a 0.8% agarose gel for the dsDNA-protein complex. The SDS-PAGE gels were stained with Coomassie blue.

**Bombardment of the plasmid DNA and repair template to rice calli**. Plasmids expressing plant codon-optimized Cas9, Cas9-VirD2, or VirD2-Cas9 and the repair templates with and without the RB were bombarded into rice (*Oryza sativa* L. ssp. *Japonica* cv. Nipponbare) calli following the protocols of Chen et al. and Shan et al. with some modification. Plasmid DNA (1 μg) and repair template (10 pM) were mixed with 0.6 μM of gold particles (50 μl of 60 μg/μl) in the presence of CaCl$_2$ (50 μl of 2.5 M CaCl$_2$ and 20 μl of 0.1 M spermidine). The gold-DNA complex was spread on the macrocarriers. Rice calli (N-80) were placed on osmotic media for 4 to 6 h and bombarded with the plasmid and repair template mixture using a Bio-Rad P1000 biolistic gun with 1100 p.s.i. rupture disks. After 24 h, the calli were spread on recovery media plates and kept in the dark for 7 days before transferring to 1$^{st}$ selection media.

**Regeneration of rice plants from the bombarded calli**. Rice plants were regenerated following the biolistic protocol of Chen et al. and Shan et al. with some modifications. Briefly, after resting the calli on recovery media for 7 days in the dark, the calli were transferred to hygromycin selection media and kept in the dark for 4 weeks. The tissues were transferred to regeneration media, and after three to four small regenerated shoots appeared, the tissues were transferred to rooting media MS salt with 0.1 g inositol, 30 g of sucrose, 2.4 g phytagel, and 30 mg hygromycin. The rooted, regenerated plants were transferred to soil in a greenhouse.

**T7EI mutation detection assay**. We performed the T7EI assay to determine the nuclease activity of Cas9-VirD2 and VirD2-Cas9. Briefly, genomic DNA was extracted from pooled calli and the target-flanking fragments were PCR amplified. After denaturation and reannealing, the PCR amplicons (200 ng) were treated with T7EI for 1 h at 37 °C. The DNA was separated on a 2% agarose gel. The frequency of Indels was calculated using ImageJ software.

**Confirmation of the MfeI recognition site insertion**. To confirm the efficient repair and the insertion of the MfeI recognition site at the targeted *ALS* locus via Cas9-VirD2 and VirD2-Cas9, we extracted genomic DNA from the pooled calli or individual plants and the target-flanked fragments were PCR amplified. The PCR products were treated with MfeI-HF restriction enzyme for 2 h at 37 °C. The DNA was separated on a 2% agarose gel. The frequency of the MfeI recognition site insertion was calculated using ImageJ software.

**Allele-specific PCR for confirmation of the repair**. To perform the allele-specific PCRs for identifying the herbicide-resistant *ALS* allele and the HA epitope tagging, we designed the insertion sequence-specific forward primers and a reverse primer complementary to the genomic sequence outside the homology arm. To detect the herbicide-resistant *ALS* allele, we performed 35 PCR cycles, while to quantify the rate of the HA epitope sequence insertion, we performed 22 PCR cycles.

**Sanger sequencing**. To confirm the exact sequence modifications at the genome level, we analyzed all the genetic modifications with Sanger sequencing. For all Sanger sequencing analyses, the PCR-amplified products of the target-flanking DNA were first cloned into the pJet2.1 cloning vector and the plasmids were sent for Sanger sequencing. The sequence modifications were confirmed by aligning the DNA reads and the corresponding chromatographs.

**Amplicon sequencing**. To confirm the exact rate of HDR at the genome level, we analyzed all the genetic modifications with amplicon sequencing. The target-flanking regions were amplified with 22 cycles of PCR, and libraries were made using the standard TruSeq platform with the Illumina protocol.

**Confirmation and heritability of herbicide resistance in rice plants**. To confirm the herbicide resistance in the regenerated plants, 10 to 20 seeds were sterilized and grown on 1/2 MS media containing the herbicide bispyribac (0.4 μM). Wild-type rice plants and plants expressing Cas9-VirD2 or VirD2-Cas9 without repair templates were used as controls. To confirm the heritability in the $T_1$ generation, seeds were grown on 1/2 MS media containing bispyribac (0.4 μM). Wild-type rice plants and plants expressing Cas9-VirD2 or VirD2-Cas9 without repair templates were used as controls. Germinated seedlings (6 days old) were used for photographs.

**Immunoprecipitation for binding ssDNA and Cas9-VirD2**. Wild type (Nipponbare) sterilized seeds were germinated in jars. Protoplasts were isolated and transfected with protoplast compatible vectors expressing Cas9, Cas9-VirD2 and Cy5-labeled ssDNA with and without RB sequence following the protocol of Butt et al.[15]. Total protein was extracted in buffer containing 100 mM Tris-Cl, pH 7.5, 150 mM NaCl, 0.5% NP-40, 1 mM EDTA, 3 mM DTT and protease inhibitors. For immunoprecipitation, anti-Flag-M2-magnetic beads (Sigma) were incubated with the total extracted protein at 4 °C for 1 h. After incubation the beads were thoroughly washed three times with 1x PBS buffer. After the last wash, 50 μl of 1x PBS was added to each sample and at incubated at 95 °C for 3 min. Half of the sample was loaded on a 6% agarose gel (Invitrogen) to separate the DNA–protein complexes and imaged for Cy5 with the gel imager, Typhoon TRIO (GE Healthcare). The other half of the sample was separated on SDS-PAGE as input for immunoblot. The immunoblot was incubated with the appropriate primary antibody (anti-Flag antibody, 1:2,000) at room temperature for two hours and anti-mouse IgG 1:3000 for 1 h (Santa Cruz Biotechnology). Immunoblots were treated with ECL (SuperSignal West Pico Plus Thermo Fisher Scientific) and the blots were detected with ImageQuant LAS4000 (GE Healthcare).

**Statistics and reproducibility**. The data presented were obtained from more than three independent repeats of multiple calli bombarded or multiple plants recovered from each experiment independently. Furthermore, the experimental results were reproduced and validated by the co-authors individually. The HDR efficiency via Cas9, VirD2-Cas9 and Cas9-VirD2 were compared in percentile.

**Reporting summary**. Further information on research design is available in the Nature Research Reporting Summary linked to this article.

## Data availability
All relevant data and clones from this work are available from the Laboratory of Genome Engineering and Synthetic Biology at KAUST upon request to corresponding author. Addgene plasmid ID numbers for newly-generated plasmids are given in the Methods.

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

## Acknowledgements

We thank members of the Laboratory for Genome Engineering and Synthetic Biology at King Abdullah University of Science and Technology for helpful discussions and comments on the manuscript. This research work is supported by King Abdullah University of Science and Technology (KAUST) baseline funding to Magdy Mahfouz.

## Author contributions

M.M. conceived the research. Z.A. and H.B. designed the research. Z.A., A.S., K.S., M.T., H.B., A.A., R.K., N.H. and A.K. performed the research. M.M., S.H., Z.A., K.S., M.T. and A.S. analyzed the data. Z.A. and M.M. wrote the paper with input from all co-authors. All authors read and approved the final manuscript.

## Competing interests

The authors declare no competing interests.
