## [Peer Review File · Communications Biology]

Reviewers' comments:

Reviewer #1 (Remarks to the Author):

Gene targeting via homology-directed repair (HDR) is difficult to achieve in plants. Despite considerable effort, there are few examples of successful systems leading to regenerated plants with targeted insertions. The approach of bringing the repair template close to the double strand break site offers the chance of increased efficiency of HDR. This could be very important for both advancing research and potentially for crop improvement. This is an area attracting considerable interest within the plant community.

In this paper, the authors demonstrate increased efficiency of targeted HDR in rice using an approach that fuses Cas9 to VirD2, therefore bringing the repair template into close proximity to the double strand break. The rates of HDR were from 8 to 23% depending on the target locus. A strength of the paper is that it demonstrated targeting of two different loci and that the increases of HDR are similar or higher than those seen with alternative approaches. The authors provide convincing arguments for the advantages offered by their system.

I would like to see minor changes before publication.

The figure legends do not contain all the necessary information to interpret the figures. For example, it is necessary to explain the abbreviations SRB and NRB.

I would like to see some consideration of the possibility of PCR artefacts affecting the results. Using primers where one is within the inserted sequence and one outside the repair template can lead to artefacts that look exactly like the expected product when sequenced. Proof of targeted HDR requires PCR using primers that are both outside of the repair template followed by sequencing. This is not possible in the pooled callus samples used in this study as there will be wildtype material present. However, the possibility of artefacts should be considered. The authors do provide full proof of targeted HDR when plants are recovered, and it is shown that the precise repair is inherited by progeny plants. Any possible PCR artefacts would probably only have a small impact on the efficiency figures.

Lines: 284-286 - The authors state: 'The genotyping by MfeI digestion (Figure 5a) confirmed a recovery of up to 16.66% with Cas9-VirD2 and up to 9.67% repaired plants with the templates containing the RB sequence compared to 1.96% for Cas9-VirD2 and 1.58% with VirD2-Cas9 with templates containing no RB sequence.'

This is not clear and needs rewriting.

Reviewer #2 (Remarks to the Author):

Ali et al describe a novel and innovative way of tethering to improve gene targeting efficiencies in plants. They express ORFs for Cas9-VirD2 fusions in rice cells in presence of a targeting vector that contains a border sequence. Due to the fusion of Cas9 to VirD2 the processed template is kept close to the homology at the moment the break is induced. In principle this a nice story and interesting to the general public, but unfortunately in its present form not yet ripe for publication. There are a row of important questions that have to be addressed before publication. In detail:

P 5 line 124:

In the light that that Polq is essential for T-DNA integration I do not think the statement that "VirD2 integrates the T-DNA into the nuclear genome" is correct.

P7 line 160 - 163

I do not think that the authors can conclude that there is no effect of the fusions on the cutting efficiency of Cas9. For this conclusion, the authors would have needed to perform a kinetic analysis of the reaction in a direct comparison with a similar amount of Cas9 protein analyzing multiple time points. For such a conclusion, a simple end point analysis is just not conclusive.

P9 line 218-244

I have tremendous problems with the way the targeting efficiencies are calculated. Obviously, the authors found calli with no gene targeting events and calli which contained GT events - but most

probably only in a few cells. So giving frequencies of 20% is a massively misleading. The interpretation of the deep sequencing data is not clear for me, either. How many reads in total from all reads showed the desired gene edits? Inky this number give us a clear picture on the GT efficiency. P11 279ff

In my understanding, it is very unlikely that due to spontaneous mutations a herbicide resistance arises – at let at much lower frequencies that GT events. Moreover, also other kind of mutation might lead spontaneous resistance, which can be discriminated by sequencing. It is no clear from me whether or not the regenerated plant were indeed clonal for the targeting events. Did all T0 plants shown in table 6B carry a heritable allele changed by GT? Did they all produce progeny? For how many of them segregation analysis was performed? Supp. figure 6 is not very informative, it would be better to show a table with a statistical evaluation.

Discussion:

I guess, the original idea of the authors was to develop a completely different strategy: Using VirD2-Cas9 fusions in *Agrobacterium* to produce a T-DNA complex that is targeted by Cas9 to the break site (As done by the Hooykaas group some time ago with I-SceI: *Sci Rep.* 2015 Feb 9;5:8345. doi: 10.1038/srep08345). Did the authors test this approach with the tools established in this study? I think it is important that the authors give the reader some information on this interesting question.

Reviewer #3 (Remarks to the Author):

This manuscript submitted by "Ali et al" tested the possibility of using a relaxase-fused Cas9 protein to facilitate precise gene editing in plants. The authors proposed that precise gene editing events can be generated and inherited efficiently from bombarded rice calli. This can be true if the targeted gene locus was ALS and herbicide-assistance screening was performed as described in Page 11. Line 282. However, in case of the HA-tag fusion, no engineered plant was reported so as to raise the concern about the editing efficiency of this system.

According to this manuscript, the authors hypothesized that virD2-fused Cas9 system have two advantages over the bare Cas9 system for directing targeted gene replacement. First, the covalent bind affinity between virD2 and RB containing ssDNA molecules enable the colocalization of donor templates with the Cas9-induced DSBs. Second, virD2 might "help the proper integration of the repair templates at the target site." These hypotheses are potentially interesting, however, neither of the two hypotheses were validated or even analyzed in the manuscript.

For the first hypothesis, previous studies have shown that "when VirD1 and VirD2 were overexpressed in ..., they remained covalently bound to the 5' end of processed ssDNA" (Page 5, line 95). But according to the manuscript, it seems that VirD1 was not required for either dsDNA nicking (Fig 2B) or the covalent binding between VirD2 and ssDNA (Fig 2C). Therefore, a strong evidence is definitely required to support their conclusions. However, it was somewhat disappointing to find that the in vitro binding assay between virD2-Cas9 and RB containing ssDNAs was too subtle to be persuasive. At least, it is not clear whether the retarded protein bands in Fig2C is due the linkage of ssDNA or other reasons. To answer this question, immunoprecipitate assay can be a good method to validate their in vivo binding.

As to the second hypothesis, it is hard to understand why virD2 can help to improve the HdR efficiency of CRISPR-induced DSBs. It is already known that the native role of VirD2 is to direct the random integration of *Agrobacterium*-derived T-DNA into plant genome. This process is accomplished via the error-prone MMEJ pathway rather than the error-free HdR pathway. Even though the virD2-Cas9 fusion were able to bring the donor template to the DSB sites, it is more reasonable to hypothesize that the proportion of imprecise DNA integration rather than precise gene replacement would increase. Moreover, considering that the donor templated were chemically-modified and introduced into rice calli via biolistic transformation, genome-wide random insertion of the modified DNAs might occur with high frequency. Nevertheless, such possibilities were not analyzed or discussed in the manuscript. In general, the concept of using virD2 to engineer the colocalization of Cas9 and RB containing ssDNAs is interesting. Nevertheless, the critical experiments were performed roughly and the conclusions were made arbitrarily. Actually, if the bombardment can be performed using ssDNA-tethered ribonuclease protein complexes with the assistance of HdR promoting elements, it is more likely that the precise

gene editing efficiency will be improved.

Other comments,

1. For the EMSA assays in Supplemental Figure 2 and 3, the indicated bands of nucleo-protein complex were not retarded prominently, even though the molecular weight of the whole complex should exceed 200Kd. Besides, the retarded bands in Figure S2 was too weak compared to the free DNA bands, suggesting that the in vitro binding system was not efficient or need to be optimized with protein/DNA content gradient.
2. For the statistic results in Fig 4B and C, data from several controlled samples were not presented, eg. Cas9-VirD2/VirD2-Cas9 + NRB/mNRB.
3. For the allelic specific PCR results in Fig 4B, it seems that only SNPs were introduced into the repair templates (Fig4A), but the resulted PCR bands were changed in size. So it is difficult to understand how the template allele can be differentiated from the wild type one?
4. For the high-throughput sequencing results in Fig 4E and Fig 5E, it is weird to use the pie charts if the sum of all components was not 100%. The authors are suggested to summarize the DSB repair outcomes of each sequencing samples according to the proportion of different mutation types, eg indels, fragment insertions, deletions, rearrangements and replacements.

Point by point reply to Reviewer's comments.

Reviewer # 1. (Remarks to the Author):

Gene targeting via homology-directed repair (HDR) is difficult to achieve in plants. Despite considerable effort, there are few examples of successful systems leading to regenerated plants with targeted insertions. The approach of bringing the repair template close to the double strand break site offers the chance of increased efficiency of HDR. This could be very important for both advancing research and potentially for crop improvement. This is an area attracting considerable interest within the plant community.

In this paper, the authors demonstrate increased efficiency of targeted HDR in rice using an approach that fuses Cas9 to VirD2, therefore bringing the repair template into close proximity to the double strand break. The rates of HDR were from 8 to 23% depending on the target locus. A strength of the paper is that it demonstrated targeting of two different loci and that the increases of HDR are similar or higher than those seen with alternative approaches. The authors provide convincing arguments for the advantages offered by their system.

Reply: We thank the reviewer for the positive and encouraging comments about our manuscript and appreciation of the idea, project design, and the importance this work to advance gene editing research and its potential application for crop improvement.

I would like to see minor changes before publication.

The figure legends do not contain all the necessary information to interpret the figures. For example, it is necessary to explain the abbreviations SRB and NRB.

Reply: We thank the reviewer for pointing this out. All the words and their abbreviations were properly addressed and included to the revised manuscript text and in the figure legends.

I would like to see some consideration of the possibility of PCR artefacts affecting the results. Using primers where one is within the inserted sequence and one outside the repair template can lead to artefacts that look exactly like the expected product when sequenced. Proof of targeted HDR requires PCR using primers that are both outside of the repair template followed by sequencing. This is not possible in the pooled callus samples used in this study as there will be wildtype material present. However, the possibility of artefacts should be considered. The authors do provide full proof of targeted HDR when plants are recovered, and it is shown that the precise repair is inherited by progeny plants. Any possible PCR artefacts would probably only have a small impact on the efficiency figures.

Reply: We thank the reviewer for this excellent comment. We agree with the reviewer about the possibility of PCR artifacts. And we thank the reviewer for understanding the complete scenario of using the allele specific PCR. These PCRs were used to confirm that our VirD2-Cas9 is active at DSBs at calli level for gene editing and also to reduce the sample number in the subsequent steps at plant level. The main goal was to recover properly engineered herbicide resistant plants and the gene editing was confirmed by using primers flanking the repair site but completely outside of the repair template sequence. The target site flanking PCR fragments were then subjected to Sanger sequencing. As discussed by the reviewer we do provide full proof of targeted HDR in the recovered plants with the suggested primers and confirmed that the precise repair was inherited into progeny plants. Also, we included another locus, *CCD7*, where we performed all the molecular analysis using primers flanking the target site.

Lines: 284-286 - The authors state: 'The genotyping by MfeI digestion (Figure 5a) confirmed a recovery of up to 16.66% with Cas9-VirD2 and up to 9.67% repaired plants with the templates containing the RB sequence compared to 1.96% for Cas9-VirD2 and 1.58% with VirD2-Cas9 with templates containing no RB sequence.'

This is not clear and needs rewriting.

Reply: We thank the reviewer for this excellent comment and for pointing this out. The respective corrections were made in the revised version of the manuscript.

Reviewer #2 (Remarks to the Author):

Ali et al describe a novel and innovative way of tethering to improve gene targeting efficiencies in plants. They express ORFs for Cas9-VirD2 fusions in rice cells in presence of a targeting vector that contains a border sequence. Due to the fusion of Cas9 to VirD2 the processed template is kept close to the homology at the moment the break is induced. In principle this a nice story and interesting to the general public, but unfortunately in its present form not yet ripe for publication. There are a row of important questions that have to be addressed before publication.

Reply: We thank the reviewer for the endorsement of our novel and innovative idea of tethering the repair template to improve gene targeting efficiencies in plants. We also thank the reviewer for the positive feedback, encouragement, and recommendations to improve the manuscript.

In detail:

P 5 line 124:

In the light that that Polq is essential for T-DNA integration I do not think the statement that “VirD2 integrates the T-DNA into the nuclear genome” is correct.

Reply: We thank the reviewer for this excellent comment. We agree with the reviewer that T-DNA integration is a multi-factor and complex process involving nicks and micro homology as prerequisites for T-DNA integration. VirD2 is the key factor that *Agrobacterium tumefaciens* uses in all steps of plant transgenesis. The multi-purpose VirD2 protein contains an endonuclease domain for nicking at RB and LB of the Ti plasmid, tyrosine 29 for covalent binding with the T-DNA, a bipartite nuclear localization sequence (NLS), and the omega domain. Mysore et. al, 1998 [1] demonstrated that Omega domain of VirD2 is important for T-DNA integration in the plant nuclear genome

and Pelczar et.al 2004 [2] demonstrated that VirD2 is sufficient for the integration of a T-DNA even into mammalian genome in the absence of other bacterial or plant factors. Likewise, Loo et.al, 2015 [3] demonstrated VirD2–histone interactions in yeast cells as a prelude to integration of T-DNA into the host chromosomal DNA. Moreover, VirD2 bound T-DNA with homology ends or VirD2-meganucleases bound T-DNA were shown to enhance gene targeting via HDR (Rolloos et.al 2015) [4].

We amended the text according to these studies and added necessary information and textual justification for the role of different factors including VirD2 in T-DNA integration into the plant genome.

1. Mysore, K.S., et al., *Role of the Agrobacterium tumefaciens VirD2 protein in T-DNA transfer and integration*. Mol Plant Microbe Interact, 1998. **11**(7): p. 668-83.
2. Pelczar, P., et al., *Agrobacterium proteins VirD2 and VirE2 mediate precise integration of synthetic T-DNA complexes in mammalian cells*. EMBO Rep, 2004. **5**(6): p. 632-7.
3. Wolterink-van Loo, S., et al., *Interaction of the Agrobacterium tumefaciens virulence protein VirD2 with histones*. Microbiology, 2015. **161**(Pt 2): p. 401-10.
4. Rolloos, M., P.J. Hooykaas, and B.J. van der Zaal, *Enhanced targeted integration mediated by translocated I-SceI during the Agrobacterium mediated transformation of yeast*. Sci Rep, 2015. **5**: p. 8345.

P7 line 160 – 163

I do not think that the authors can conclude that there is no effect of the fusions on the cutting efficiency of Cas9. For this conclusion, the authors would have needed to perform a kinetic analysis of the reaction in a direct comparison with a similar amount of Cas9 protein analyzing multiple time points. For such a conclusion, a simple end point analysis is just not conclusive.

Reply: We thank the reviewer for this important comment. We agree with the reviewer on this point that a simple end point analysis is not conclusive. We amended the text and stated that VirD2-Cas9 fusions still retain a high nuclease activity of Cas9 that would be sufficient for the cleavage of the target locus.

P9 line 218-244

I have tremendous problems with the way the targeting efficiencies are calculated. Obviously, the authors found calli with no gene targeting events and calli which contained GT events - but most probably only in a few cells. So giving frequencies of

20% is a massively misleading. The interpretation of the deep sequencing data is not clear for me, either. How many reads in total from all reads showed the desired gene edits? Inky this number give us a clear picture on the GT efficiency.

Reply: We thank the reviewer for this excellent comment. We want to clarify this issue to the reviewer and the reader. We conducted allele-specific PCR to determine how many calli carried repair events. According to our experiments, 20% of the calli showed positive allele-specific PCR indicating the presence of the repair event. Furthermore, we conducted deep amplicon sequencing to validate our allele-specific PCR results and determine the approximate number of calli we needed to proceed for regeneration and recovery of a reasonable number of plants harboring the edited allele. Subsequently, we compared frequency of the presence of the repaired allele using Cas9, Cas9.VirD2, and VirD2.Cas9. Our data showed that both Cas9.VirD2 and VirD2.Cas9 enhanced the gene editing frequency by several fold compared to Cas9 only.

P11 279ff

In my understanding, it is very unlikely that due to spontaneous mutations a herbicide resistance arises – at let at much lower frequencies that GT events. Moreover, also other kind of mutation might lead spontaneous resistance, which can be discriminated by sequencing. It is no clear from me whether or not the regenerated plant were indeed clonal for the targeting events.

Reply: We thank the reviewer for this excellent comment. We agree with the reviewer that in rare possibilities some spontaneous mutations might occur resulting in herbicide resistance. Our genotyping data show that only the resistant calli possessing a repaired allele and the regenerated plantlets conferring herbicide resistance contained the repaired allele. Moreover, the progeny of these plants contained the repaired allele indicating that these repair events were clonal and heritable. Please note that we added extra sequence modifications in the repair template to distinguish between true edits resulting from our VirD2.Cas9 machinery and other types of mutation. Our data show

that the resistant calli and plantlets possessed the exact repair template. Furthermore, the heritability of gene edits and herbicide resistance in seed progeny rule out that such observed resistance resulted from tissue culture somaclonal variation. These data provide strong evidence on the usefulness and utility of this system for targeted gene editing.

Did all T0 plants shown in table 6B carry a heritable allele changed by GT?

Reply: We thank the reviewer for this comment. We checked the heritability of the repaired allele in seed progeny in two lines (line 70 and Line 121). Both lines showed proper heritability of the repaired allele and herbicide resistance phenotype (Supplementary Figure 8).

Did they all produce progeny? For how many of them segregation analysis was performed? Supp. figure 6 is not very informative, it would be better to show a table with a statistical evaluation.

Reply: We thank the reviewer for this suggestion. We performed the segregation analysis for two lines (line 70 and line 121). These lines showed the proper segregation of the repaired allele and herbicide resistance phenotype. A supplementary table was added to the manuscript for details.

Discussion:

I guess, the original idea of the authors was to develop a completely different strategy: Using VirD2-Cas9 fusions in Agrobacterium to produce a T-DNA complex that is targeted by Cas9 to the break site (As done by the Hooykaas group some time ago with I-SceI: Sci Rep. 2015 Feb 9;5:8345. doi: 10.1038/srep08345). Did the authors test this approach with the tools established in this study? I think it is important that the authors give the reader some information on this interesting question.

Reply: We thank the reviewer for this excellent comment. We are already developing

this system for Cas9.VirD2 translocation and delivery into plant cells via *Agrobacterium*. We have encouraging results indicating that this system may be useful for plant genome engineering applications. We have included text in the discussion section, in the revised version of the manuscript to give the reader information on these possibilities.

Reviewer #3 (Remarks to the Author):

This manuscript submitted by “Ali et al” tested the possibility of using a relaxase-fused Cas9 protein to facilitate precise gene editing in plants. The authors proposed that precise gene editing events can be generated and inherited efficiently from bombarded rice calli. This can be true if the targeted gene locus was ALS and herbicide-assistant screening was performed as described in Page 11. Line 282.

Reply: We thank the reviewer for the positive feedback, encouragement, and the endorsement of our idea of tethering the repair template to DSBs to improve gene targeting efficiencies in plants.

However, in case of the HA-tag fusion, no engineered plant was reported so as to raise the concern about the editing efficiency of this system.

Reply: We thank the reviewer for this excellent comment. We understand that only one target might not be sufficient to ensure the generalizability of the system for gene editing applications. We repeated the experiment for *HDT* and can confirm the presence of repair events in calli. Furthermore, we employed this strategy to edit the *CCD7* gene and our data show that calli possessed the repaired allele. We have recovered plants from these calli carrying the edited *CCD7* allele indicating the generalizability of this approach for targeted gene editing in plants. The results of the new experiments (Figure 8, supplementary Figure 10 and 11) were added to the revised version of the manuscript.

According to this manuscript, the authors hypothesized that virD2-fused Cas9 system have two advantages over the bare Cas9 system for directing targeted gene replacement. First, the covalent bind affinity between virD2 and RB containing ssDNA molecules enable the colocalization of donor templates with the Cas9-induced DSBs. Second, virD2 might “help the proper integration of the repair templates at the target site.” These hypotheses are potentially interesting, however, neither of the two hypotheses were validated or even analyzed in the manuscript.

Reply: We thank the reviewer for taking a keen interest and careful reading of our manuscript. VirD2 is the key factor that *Agrobacterium tumefaciens* uses in all steps of plant transgenesis. Mysore et. al, 1998 demonstrated that the Omega domain of VirD2 is important for T-DNA integration into the plant nuclear genome and Pelczar et.al 2004 [2] demonstrated that VirD2 is sufficient for the integration of a T-DNA even into the mammalian genome in the absence of other bacterial or plant factors. Likewise, Loo et.al, 2015 [3] demonstrated the VirD2 and histone interactions in yeast cells as a prelude to integration of T-DNA into the host chromosomal DNA. Moreover, VirD2 bound T-DNA with homology ends or VirD2-meganucleases bound T-DNA enhanced gene targeting via HDR (Rolloos et.al 2015) [4]. We have included these citations and studies in the revised version of the manuscript.

Our developed protein/DNA binding assay clearly demonstrates the covalent binding of the RB containing ssDNA to the VirD2-Cas9 fusion. Additionally, as recommended by the reviewer, we conducted a new experiment to confirm and validate the binding activity of VirD2-Cas9 fusion to the repair templates both *in-vitro* and in rice protoplasts (Figure 2d and 3a). The results of these experiments support the main idea of this manuscript that the Cas9.VirD2 system enhances the precise gene targeting and the recovery of edited plants.

For the first hypothesis, previous studies have shown that “when VirD1 and VirD2 were overexpressed in, they remained covalently bound to the 5’ end of processed ssDNA” (Page 5, line 95). But according to the manuscript, it seems that VirD1 was not required for either dsDNA nicking (Fig 2B) or the covalent binding between VirD2 and

ssDNA (Fig 2C). Therefore, a strong evidence is definitely required to support their conclusions.

Reply: We thank the reviewer for this excellent comment and for pointing this important issue out. Yes, VirD2 alone is sufficient for nicking and covalent binding to RB containing ssDNA. Other studies have clearly demonstrated the absolute ability of the VirD2 to nick and covalently bind to the RB containing ssDNA without any addition of VirD1 (Pansegrau et.al, 1992 [5] and Bernardinelli et.al, 2017 [6]). These citations were added to the revised version of the manuscript. Furthermore, we tested this in our lab and we show that VirD2 alone is capable of producing nicks and covalent binding to ssDNA (unpublished results). It is worth noting that VirD2 is a multi-purpose protein that contains an endonuclease domain for nicking at RB and LB of Ti plasmid, tyrosine 29 for covalent binding with nicked T-DNA, bipartite nuclear localization sequence (NLS) and the omega domain for helping in the integration of T-DNA into the plant genome.

However, it was somewhat disappointing to find that the *in vitro* binding assay between virD2-Cas9 and RB containing ssDNAs was too subtle to be persuasive. At least, it is not clear whether the retarded protein bands in Fig2C is due the linkage of ssDNA or other reasons. To answer this question, immunoprecipitation assay can be a good method to validate their *in vivo* binding.

Reply: We thank the reviewer for this excellent comment and for suggesting immunoprecipitation assays to confirm the binding in the cellular environment. Our developed protein/DNA binding assay is a clear demonstration of VirD2 *in vitro* activity. VirD2 has the innate ability to re-ligate the ssDNA nicked products available in the *in vitro* reaction and the reaction appears to saturate at ~40% [5]. Such re-ligation is compensated by integration of the T-DNA into the plant genome. Phosphorothioate end modification and use of exonuclease to remove the unprotected nicked substrate from the reaction were previously shown to enhance the rate of irreversible conjugation (VirD2 with the RB containing ssDNA) by lowering the dynamic conjugation equilibrium [6]. However, to remove any doubt, we performed two independent assays to confirm the binding of the repair templates to VirD2-Cas9 *in-vitro* and immunoprecipitation of the repair template from rice protoplasts. The phosphonothioate end modified repair

templates with RB (mT-RB) were incubated with VirD2-Cas9 fusion and reaction was carried out in presence of Exonuclease1. The removal of the nicked unbound 3'-open ssDNA from the reaction by Exonuclease1 (3' - 5' nuclease activity) led to the complete binding of the repair templates to VirD2-Cas9 (Figure 2d) as demonstrated by gel mobility shift assay. Moreover, as suggested by the reviewer, we cloned the VirD2-Cas9 into rice protoplast compatible vectors and performed the immunoprecipitation-based pull-down of the Cy5-labeled RB containing repair template in rice protoplast. Our results (Figure 3a) clearly demonstrate the efficient binding of the repair templates to the VirD2-Cas9.

As to the second hypothesis, it is hard to understand why virD2 can help to improve the HDR efficiency of CRISPR-induced DSBs. It is already known that the native role of VirD2 is to direct the random integration of *Agrobacterium*-derived T-DNA into plant genome. This process is accomplished via the error-prone MMEJ pathway rather than the error-free HDR pathway. Even though the virD2-Cas9 fusion were able to bring the donor template to the DSB sites, it is more reasonable to hypothesize that the proportion of imprecise DNA integration rather than precise gene replacement would increase.

Reply: We agree with the reviewer that according to one model of *Agrobacterium* mediated genetic transformation of plants the process of T-DNA integration into the host genome is accomplished via the MMEJ pathway. But first it is only in natural genetic transformation where DSBs are not present, targeted, or predominant. Second it is dependent on the randomly available nicks in the host genomic DNA. Third in the natural system normally homologous ends are not available. Fourth it was shown, if available or generated through site-specific nucleases, that T-DNA prefers to integrate into the DNA double-strand break sites (Salomon and Puchta, 1998, [6] Tizfira et. al 2003 [7]). Fifth VirD2 bound T-DNA with homology at the ends or VirD2-meganucleases bound T-DNA were shown to enhance gene targeting via HDR (Rolloos et.al 2015). We have added these citations in the revised version of the manuscript to provide textual justification for this point.

Moreover, considering that the donor templated were chemically-modified and introduced into rice calli via biolistic transformation, genome-wide random insertion of the modified DNAs might occur with high frequency. Nevertheless, such possibilities were not analyzed or discussed in the manuscript.

Reply: We thank the reviewer for pointing this out. This possibility is discussed in the revised version of the manuscript and certainly this point will be addressed in detail in our future work.

In general, the concept of using virD2 to engineer the colocalization of Cas9 and RB containing ssDNAs is interesting. Nevertheless, the critical experiments were performed roughly and the conclusions were made arbitrarily. Actually, if the bombardment can be performed using with the assistance of HdR promoting elements, it is more likely that the precise gene editing efficiency will be improved.

Reply: We thank the reviewer for this excellent comment and for the ideas to improve the efficiency by using HDR promoting elements. Our group is already working on this aspect and this point will be fully addressed in our future work. In this revised version of the manuscript we discussed this suggested possibility.

Other comments,

1. For the EMSA assays in Supplemental Figure 2 and 3, the indicated bands of nucleoprotein complex were not retarded prominently, even though the molecular weight of the whole complex should exceed 200Kd. Besides, the retarded bands in Figure S2 was too weak compared to the free DNA bands, suggesting that the *in vitro* binding system was not efficient or need to be optimized with protein/DNA content gradient.

Reply: We thank the reviewer for pointing out this important issue and suggesting immunoprecipitation assay to confirm the binding in the cellular environment. We discussed this point above. Our developed protein/DNA binding assay is a clear demonstration of VirD2 *in vitro* activity. VirD2 has the innate ability to re-ligate the ssDNA nicked products available in the *in-vitro* reaction and the reaction appears to saturate at ~40% [5]. Such re-ligation is compensated by integration of the T-DNA into

the plant genome. Phosphorothioate end modification and use of Exonuclease to remove the unprotected nicked substrate from the reaction were previously shown to enhance the rate of irreversible conjugation (VirD2 with the RB containing ssDNA) by lowering the dynamic conjugation equilibrium [6]. However, to remove any doubt, we performed two independent assays to confirm the binding of the repair templates to VirD2-Cas9 fusion *in-vitro* and immunoprecipitation of the repair template from rice protoplasts. The Phosphonothioate end modified repair templates with RB (mT-RB) were incubated with VirD2-Cas9 fusion and reaction was carried out in presence of Exonuclease 1 (for 3' - 5' nuclease activity). The removal of the nicked unbound 3'-open ssDNA from the reaction by Exonuclease 1 led to the complete binding of the repair templates to VirD2-Cas9 (Figure 2d) as demonstrated by gel mobility shift assay. Moreover, as suggested by the reviewer, we cloned the VirD2-Cas9 to rice protoplast compatible vectors and performed the immunoprecipitation-based pull-down of the Cy5-labeled RB containing repair template in rice protoplast. Our results (Figure 3a) clearly demonstrate the efficient binding of the repair templates to the VirD2-Cas9.

2. For the statistic results in Fig 4B and C, data from several controlled samples were not presented, eg. Cas9-VirD2/VirD2-Cas9 + NRB/mNRB.

Reply: We thank the reviewer for pointing out these missing controls. The suggested controls were added to the Fig 4 B and C in this revised version of the manuscript.

3. For the allelic specific PCR results in Fig 4B, it seems that only SNPs were introduced into the repair templates (Fig4A), but the resulted PCR bands were changed in size. So it is difficult to understand how the template allele can be differentiated from the wild type one?

Reply: We thank the reviewer for raising this point. In the allelic-specific PCR results in Fig 4B, the lower band is common (Nonspecific) in all calli including wild type. The allele

specific amplification is indicated by asterisks and arrow heads. The respective information is added to the figure legends.

4. For the high-throughput sequencing results in Fig 4E and Fig 5E, it is weird to use the pie charts if the sum of all components was not 100%. The authors are suggested to summarize the DSB repair outcomes of each sequencing samples according to the proportion of different mutation types, eg indels, fragment insertions, deletions, rearrangements and replacements.

Reply: We thank the reviewer for raising this point. We amended these figures in the revised version of the manuscript and the respective information is added to the figures and to the figure legends.

1. Mysore, K.S., et al., *Role of the Agrobacterium tumefaciens VirD2 protein in T-DNA transfer and integration*. Mol Plant Microbe Interact, 1998. **11**(7): p. 668-83.
2. Pelczar, P., et al., *Agrobacterium proteins VirD2 and VirE2 mediate precise integration of synthetic T-DNA complexes in mammalian cells*. EMBO Rep, 2004. **5**(6): p. 632-7.
3. Wolterink-van Loo, S., et al., *Interaction of the Agrobacterium tumefaciens virulence protein VirD2 with histones*. Microbiology, 2015. **161**(Pt 2): p. 401-10.
4. Rolloos, M., P.J. Hooykaas, and B.J. van der Zaal, *Enhanced targeted integration mediated by translocated I-SceI during the Agrobacterium mediated transformation of yeast*. Sci Rep, 2015. **5**: p. 8345.
5. Pansegrau, W., et al., *Site-specific cleavage and joining of single-stranded DNA by VirD2 protein of Agrobacterium tumefaciens Ti plasmids: analogy to bacterial conjugation*. Proc Natl Acad Sci U S A, 1993. **90**(24): p. 11538-42.
6. Bernardinelli, G. and B. Hogberg, *Entirely enzymatic nanofabrication of DNA-protein conjugates*. Nucleic Acids Res, 2017. **45**(18): p. e160.
7. Tzfira, T., et al., *Site-specific integration of Agrobacterium tumefaciens T-DNA via double-stranded intermediates*. Plant Physiol, 2003. **133**(3): p. 1011-23.

REVIEWERS' COMMENTS:

Reviewer #1 (Remarks to the Author):

This manuscript describes a method to enhance homology directed repair (HDR) by using a Cas9-VirD2 fusion. A huge amount of previous work has been devoted to improving HDR in plants with only limited success and low efficiencies. This manuscript, therefore, represents an important step towards improved HDR efficiencies and as such should be published.

The authors have made substantial revisions to the original manuscript and have responded very thoroughly to all the reviewers comments. Additional data has been included in supplementary figures. Of particular importance is the inclusion of data on engineering plant architecture by targeting the CCD7 gene in rice. Adding this additional example really strengthens the paper as a clear phenotype is seen in the engineered plants and there are now three examples successfully demonstrating the benefits of using the Cas9-VirD2 fusion system. I strongly recommend publication in the current form.

Reviewer #2 (Remarks to the Author):

I read the revised version of the manuscript with interest. The authors were able to address the points that I raised and supply us with an improved manuscript. I also highly appreciate that they included further experimental data, especially the targeting of the CCD7 gene. This is a nice demonstration that the technology is useful which definitely raises the importance of the manuscript.

Reviewer #3 (Remarks to the Author):

It is good to see that the resubmitted manuscript was revised in many aspects and also included some new data to support their conclusions. However, the current version is still not very persuasive because some fundamental problems remain unsolved.

First of all, the expected optimization of protein/DNA gradient for their in vitro binding was not performed in Figure 2C. This experiment is important because it helps to estimate the binding affinity of virD2-Cas9 fusion protein to ssDNAs. Under current conditions, it seems that the relative intensity of retarded protein bands to non-retarded ones was quite low in Figure 2C and even lower in Figure 3A (anti-flag IB), which suggested that the majority of the fusion proteins were unable to bind a ssDNA either in vitro or in vivo. The authors argued that virD2 alone is enough for in vitro binding of ssDNA. But it seems that the efficiency can be pretty low due to many reasons. If this problem is not fixed, how can we expect the prominent improvement of targeted DNA integration into transformed cells? In regard to the immunoprecipitation assay in Figure 3A, it's unreasonable to load the IP products into 6% Agarose gel and SDS-PAGE separately. Since the ssDNAs were already labeled with Cy5, it's more informative to visualize the DNA band and the immunoblot band of fusion protein from the same acrylamide gel (Reference 36 of this manuscript). And only in this way can they validate the covalent binding of ssDNAs to fusion proteins and determine the proportion of ssDNA-protein complexes to nascent DNA or proteins.

Another problem is that the statistics of this manuscript can be completely misleading if the frequency of those imprecise repair outcomes were not analyzed. According to (Tsai, et al. 2015), 5' and 3' phosphorothioated dsODNs can inserted into the induced DSB sites via NHEJ pathway with extremely high efficiency. Since it is still not clear whether the modified ssODNs will have the same effect as dsODNs, it is necessary to evaluate the proportion of precise editing events by using an accurate and high throughput genotyping method, e.g. deep sequencing for flanking PCR amplicons.

However, the currently used allele specific and flanking PCR methods cannot completely discriminate the precise editing events from those imprecise ones, especially for those containing mismatches at the junctions. Moreover, the accuracy of repair junctions cannot be judged from the provided chromatographs, even though Sanger sequencing were performed in Figure 5D, 6D and 7G. So, it is hard to estimate to which extent the statistics for precise gene editing events are reliable.

Actually, it is hard to understand why the colocalization of donor templates to DSB sites can improve

precise gene editing. Theoretically, this strategy can increase the possibility of donor DNA integration into the target loci, but it won't significantly change the ratio of precise editing events to imprecise ones (which contain mismatches at the integration junctions). The authors argued that quite a lot previous studies have shown that "T-DNA prefers to integrate into the DNA double-strand break sites". This is true if the imprecise editing events were also included for analysis or the precise ones were subjected to certain kinds of selection. Nevertheless, in this manuscript, the authors reported that improvement of precise genome engineering events in absence of any selection. These results are not persuasive enough unless the proportion of different editing outcomes can be analyzed and summarized in a nonbiased way. Besides, the authors need to read their citation (Rolloos, et al. 2015) more carefully to make it clear whether the fusion of VirD2-I-SceI can improve HdR or not. In fact, this protein fusion failed to improve the HdR efficiency as expected, probably because the virD2 protein was very vulnerable when fused to other protein moieties.

The last problem is about the HdR improvement effect of protein/DNA tethering strategy compared to DNA phosphorothioation. It seems that DNA phosphorothioation had more promising effect than protein/DNA tethering in promoting targeted DNA insertion/replacement. In Figure 4C, the sample combinations, which use modified repair templates, the targeted DNA integration ratio is universally higher than those using unmodified ones. While for those using unmodified ones, no matter if RB were included in the ssDNA templates or not, Cas9-virD2 produced no expected editing at all. This result indicated from another aspect that the current protein fusion strategy is still evolving and definitely requires further optimization.

Other comments,

1. In Figure 2C, 2D, 3A, 4D, 5B and 7B, molecular markers are missing or not labeled.
2. In Figure 2B, the increasing gradient of protein/DNA molar ratio are not labeled out.
3. In Figure 2E and F, to make the results comparable, the in vitro cleavage assay for fusion proteins should be performed together.
4. In Figure 4B, the appearance of such a bright nonspecific band should be avoid. The authors can perform flanking PCR for primary screening as what they have done in Figure 8B.
5. In Figure 5D, 6D, 7C and 8G, the sequencing chromatographs should include both junctions of the integrated DNA sequences.
6. In Figure 6B, please specify which kinds of repair events are designated as "partially repaired"? Please also put these "partially repaired events" into analysis in Figure 8.

References

- Rolloos M, Hooykaas PJ, van der Zaal BJ (2015) Enhanced targeted integration mediated by translocated I-SceI during the Agrobacterium mediated transformation of yeast. *Scientific reports* 5:8345
- Tsai SQ, Zheng Z, Nguyen NT, Liebers M, Topkar VV, Thapar V, Wyvekens N, Khayter C, Iafrate AJ, Le LP, Aryee MJ, Joung JK (2015) GUIDE-seq enables genome-wide profiling of off-target cleavage by CRISPR-Cas nucleases. *Nature biotechnology* 33:187-197

Dear Dr Morneau,

Thank you for your interest, positive and encouraging feedback on our revised manuscript. We are glad to hear the positive feedback from the reviewers, especially reviewer 1 and Reviewer 2.

Our study presents the first evidence on the applicability of the use of VirD2-Cas9 fusions to achieve efficient precise genome engineering in plants. Our data demonstrated that VirD2-Cas9 covalently bound to the RB-containing DNA repair templates and tethered the repair template in close proximity to the DSB, resulting in an enhanced rate of gene editing in rice. As evident from the molecular assays and sequencing data, we achieved efficient gene editing and recovered precisely repaired plants at three different targeted loci. With the successful engineering of *CCD7* and *ALS*, we recovered the desired multi-tiller/short-stature and herbicide resistant rice plants, respectively. Additionally, with our VirD2-Cas9 gene editing system, we engineered an in-frame HA tag into the endogenous *HDT* gene in the rice genome. These results show the multipurpose utility of the VirD2-Cas9 gene editing system for diverse genome engineering applications.

Regarding the comments from the reviewer #3, even though we included all experimental results and textual recommendations previously suggested by reviewer #3 in to the revised manuscript, still, we are more than happy to clarify any new point raised by reviewer #3.

We are confident about our research and the validity of this approach and believe that our point by point reply to these new comments should be satisfactory to reviewer #3.

Here, we would like to thank you for your interest and consideration of our manuscript for publication after addressing these minor comments. We also thank the reviewers for their time and constructive suggestions to improve our manuscript. Should you have any questions concerning the revised version of the manuscript please let me know.

We look forward to hearing from you at your earliest convenience.

With very best wishes,

Magdy Mahfouz

Reviewer #3 report:

It is good to see that the resubmitted manuscript was revised in many aspects and also included some new data to support their conclusions. However, the current version is still not very persuasive because some fundamental problems remain unsolved.

Reply: We thank the reviewer for the positive feedback on the improvement of our manuscript and on the endorsement that the new results included in our revised manuscript strengthen our idea and final conclusions.

First of all, the expected optimization of protein/DNA gradient for their *in vitro* binding was not performed in Figure 2C. This experiment is important because it helps to estimate the binding affinity of virD2-Cas9 fusion protein to ssDNAs. Under current conditions, it seems that the relative intensity of retarded protein bands to non-retarded ones was quite low in Figure 2C.

Reply: We thank the reviewer for bringing this comment back into the discussion. As suggested by the reviewer#3, we optimized the binding in Figure 2d (as an extension of Figure 2c), may be the reviewer missed the details of Figure 2d in the revised version of the manuscript.

As discussed previously, VirD2 has the innate ability to re-ligate the ssDNA nicked products available in the *in vitro* reaction and the reaction appears to saturate at ~40% [5]. In such scenario gradient increase/decrease of the Cas9-VirD2 or RB containing repair template did not help (data not shown).

To confirm/optimize the Cas9-VirD2 binding to the repair template, the direct *in-vitro* binding assay of phosphonothioate end modified repair templates with RB (mT-RB) and VirD2-Cas9 in presence of Exonuclease1 (3' - 5' nuclease activity) was applied. The removal of VirD2 (Cas9-VirD2) nicked unprotected substrate from the reaction enhanced the rate of irreversible conjugation (VirD2 with the RB containing ssDNA) and led to the complete binding of the repair templates to VirD2-Cas9 and is clearly demonstrated by gel mobility retardation of the DNA-VirD2-Cas9 complex (Figure 2d).

and even lower in Figure 3A (anti-flag IB), which suggested that the majority of the fusion proteins were unable to bind a ssDNA either *in vitro* or *in vivo*.

Reply: We thank the reviewer for the discussion on this point. Initially this experiment was suggested by the reviewer#3 just to demonstrate and confirm the binding of the repair template to the *in-planta* produced Cas9-VirD.

In a living cell, like rice protoplast, the scenario of binding is totally different than in *in-vitro* assays. In these experiments, a fixed amount of Cy5-labeled RB containing repair template is transfected to the plant protoplasts along with a plasmid harboring a Cas9-VirD2 expression cassette. Inside the plant cell *pCaMV35S* based constitutive expression of Cas9-VirD2 would produce a surplus amount of protein compared to the fixed amount of Cy5-labeled RB template provided. Moreover, in the cellular environment complete binding or 100 % activity is not possible or even may not be obligatory.

Our results (Figure 3a) clearly demonstrated the production of an active Cas9-VirD2 that can covalently bind to the repair template and it was possible to pull-down the complex via immunoprecipitation.

The authors argued that virD2 alone is enough for *in vitro* binding of ssDNA. But it seems that the efficiency can be pretty low due to many reasons. If this problem is not fixed, how can we expect the prominent improvement of targeted DNA integration into transformed cells?

Reply: On this point we don't agree with the reviewer, we practically used VirD2 in a broad range of experiments (data not shown). We observed the ability of VirD2 to bind efficiently to ssDNA. This argues against the need for any other helper protein to enhance cutting or covalent binding of the VirD2 to the single stranded RB containing substrates. We would like to mention here, that it is not the lower efficiency of the VirD2, but it is the digestion and re-ligation ability of VirD2 to keep a dynamic conjugation equilibrium of the RB containing substrate in an *in-vitro* reaction. Equilibrium shift, i.e. removal of the one part of the substrate via exonuclease leads to complete binding of the RB containing substrate to the Cas9-VirD2 (Figure 2d). Moreover, Bernardinelli *et al* 2017 demonstrated that only the first 204 amino acid of the VirD2 are sufficient for the enzymatic activity, cutting and covalent binding, to the single stranded RB containing substrates.

In regard to the immunoprecipitation assay in Figure 3A, it's unreasonable to load the IP products into 6% Agarose gel and SDS-PAGE separately. Since the ssDNAs were already labeled with Cy5, it's more informative to visualize the DNA band and the immunoblot band of fusion protein from the same acrylamide gel (Reference 36 of this manuscript). And only in this way can they validate the covalent binding of ssDNAs to fusion proteins and determine the proportion of ssDNA-protein complexes to nascent DNA or proteins.

Reply: We thank the reviewer for discussing this point. As mentioned earlier, this experiment was suggested by the reviewer to demonstrate and confirm the binding of the repair template to the in-planta produced Cas9-VirD. In the Reference 36 of this manuscript, the system used is completely different from the Immunoprecipitation assay. Bernardinelli *et al* 2017 used purified protein for complexation with the labeled RB containing substrates and the same acrylamide gel was used to detect protein and DNA. In contrast to purified proteins from bacterial cells, the amount of the proteins or DNA-protein complex pull-down with anti-body from plant protoplasts is much lower and below the detection level of Coomassie staining. There are multiple ways for the demonstration of protein-DNA complexation, but detection of the protein with antibody is

more specific than any other system. Moreover, the Flag antibody was able to detect both the Cas9-VirD2 and the shifted Cas9-VirD2-DNA complex (Figure 3a, lower Flag immunoblot panel) validated directly the binding of the in-planta produced Cas9-VirD2 to RB containing templates.

Another problem is that the statistics of this manuscript can be completely misleading if the frequency of those imprecise repair outcomes were not analyzed. According to (Tsai, et al. 2015), 5' and 3' phosphorothioated dsODNs can be inserted into the induced DSB sites via NHEJ pathway with extremely high efficiency. Since it is still not clear whether the modified ssODNs will have the same effect as dsODNs, it is necessary to evaluate the proportion of precise editing events by using an accurate and high throughput genotyping method, e.g. deep sequencing for flanking PCR amplicons.

Reply: We thank the reviewer for discussing this point. We would like to mention that Tsai, et al. 2015 used double stranded DNA (dsODNs), modified at both ends, as a suitable substrate for insertion at DSB via NHEJ in the genome of human cell lines. However, in contrast to this work, in our experiment we used ssDNA modified at both ends, but one modified end is removed and replaced by VirD2 covalent binding to the repair template. Second compared to dsDNA, ssDNA is the preferable substrate for HDR machinery, not NHEJ (Richardson *et al* 2016). HDR is a rare event in plant cells, the main objective of this study is to enhance the HDR rate in plants, and our results clearly demonstrated that bringing the ssDNA repair template to close proximity to the DSB via Cas9-VirD2 enhanced the rate of HDR in plants.

Richardson C.D., Ray G.J., DeWitt M.A., Curie G.L., Corn J.E. Enhancing homology-directed genome editing by catalytically active and inactive CRISPR-Cas9 using asymmetric donor DNA. *Nat. Biotechnol.* 2016;34:339–344.

However, the currently used allele specific and flanking PCR methods cannot completely discriminate the precise editing events from those imprecise ones, especially for those containing mismatches at the junctions. Moreover, the accuracy of repair junctions cannot be judged from the provided chromatographs, even though Sanger sequencing were performed in Figure 5D, 6D and 7G. So, it is hard to estimate to which extent the statistics for precise gene editing events are reliable.

Reply: We thank the reviewer for discussing this point. However, we would like to mention that the Sanger sequencing clearly demonstrate the efficient HDR, not the imprecise insertion. We observed only the precise changes included in the repair templates at particular sites and there are no extra sequence changes at the junctions as demonstrated in the Figure 5D. Also, we added Sanger chromatograms (including junctions) of 7G as supplementary Figure 12 indicating the precise insertion of the repair template into the genome of these plants.

Actually, it is hard to understand why the colocalization of donor templates to DSB sites can improve precise gene editing. Theoretically, this strategy can increase the possibility of donor DNA integration into the target loci, but it won't significantly change the ratio of precise editing events to imprecise ones (which contain mismatches at the integration junctions). The authors argued that quite a lot previous studies have shown that "T-DNA prefers to integrate into the DNA double-strand break sites".

Reply: As discussed previously, we agree with the reviewer that according to one model of Agrobacterium mediated genetic transformation of plants the process of T-DNA integration into the host genome is accomplished via the MMEJ pathway. But, first, it is only in natural genetic transformation where DSBs are not present, targeted, or predominant. Second, it is dependent on the randomly available nicks in the host genomic DNA. Third, in the natural system normally homologous ends are not available. Fourth, it was shown, if available or generated through site-specific nucleases, that T-DNA prefers to integrate into the DNA double-strand break sites (Salomon and Puchta, 1998, Tizfira et. al 2003). Fifth, VirD2 bound T-DNA with homology at the ends or VirD2-

meganucleases bound T-DNA were shown for gene targeting via HDR. As per this work, the discussion included “*Such methodology of “protein therapy” in combination with a gene targeting construct has the advantage that no extra nucleic acids need to be introduced besides the gene targeting construct (Rolloos et.al 2015)*” We have already added these citations in the revised version of the manuscript to provide textual justification for this point.

This is true if the imprecise editing events were also included for analysis or the precise ones were subjected to certain kinds of selection. Nevertheless, in this manuscript, the authors reported that improvement of precise genome engineering events in absence of any selection. These results are not persuasive enough unless the proportion of different editing outcomes can be analyzed and summarized in a nonbiased way.

Reply: We thank the reviewer for bringing this new point to the discussion. We clearly mentioned “ALS is involved in branched-chain amino acid biosynthesis line 189, page 8” Therefore, as a member of the amino acid biosynthetic pathway ALS is biologically essential and the failure to correctly repair the ALS locus may exert a biological pressure.

Second “only targeted conversions (W548L and S627I) can render rice plants resistant to the herbicide Bispyribac Sun, Y., *et al* 2016, line 190, page 8” which means an imprecise insertion will not be tolerated biologically, as it will impair the production of an active ALS polypeptide.

Please note that, we did not use Bispyribac-containing regeneration media in order to avoid herbicide resistance derived from spontaneous mutations via chemical (Bispyribac) pressure. To avoid this rare possibility, we regenerated the rice calli on hygromycin-containing media, as the T-DNA harbours the hygromycin resistance gene.

Besides, the authors need to read their citation (Rolloos, et al. 2015) more carefully to make it clear whether the fusion of VirD2-I-SceI can improve HdR or not. In fact, this protein fusion failed to improve the HdR efficiency as expected, probably because the virD2 protein was very vulnerable when fused to other protein moieties.

Reply: We thank the reviewer for bringing Rolloos *et al* 2015 work to the discussion. Here, we would like to mention that, Rolloos *et al* applied a totally different approach i.e. translocation of the T-DNA-VirD2-I-SceI complex from *Agrobacterium* for HDR in the **yeast cells**. The translocation of such a big complex (T-DNA-VirD2-I-SceI) specifically in the heterologous system of yeast (T4SS is mainly working for plants) is expected to be low.

Rolloos *et al* explained *“the chimerical VirD2 proteins can still support enzyme activity for DNA metabolism as shown for the ability of NLS-VirD2-I-Cre-VirFCT to support in vivo Cre-mediated recombination on a yeast target locus”*

And Rolloos *et al* 2015 discussed two reasons 1. *“the observed low transformation efficiency could be due to reduced T4SS mediated translocation of NLS-VirD2-I-SceI-VirF^{CT} compared to WT VirD2 or 2. to reduced relaxase activity of VirD2 when fused to other protein moieties. Both issues would lead to less T-strands being transferred, resulting into lower transformation efficiencies”*

In contrast to this work, we applied a completely different and direct experimental approach to bypass the limitation of T-DNA-VirD2-Cas9 complex translocation via T4SS i.e. in our applied system T-DNA-VirD2-Cas9 complex is made inside the plant cell for HDR in the same cells. Our results clearly demonstrated that, when produced inside the same cell VirD2-Cas9 fusion is active for their respective nuclease activities and enhanced precise gene editing.

The last problem is about the HDR improvement effect of protein/DNA tethering strategy compared to DNA phosphorothioation. It seems that DNA phosphorothioation had more promising effect than protein/DNA tethering in promoting targeted DNA insertion/replacement. In Figure 4C, the sample combinations, which use modified repair templates, the targeted DNA integration ratio is universally higher than those using unmodified ones. While for those using unmodified ones, no matter if RB were included in the ssDNA templates or not, Cas9-virD2 produced no expected editing at all.

This result indicated from another aspect that the current protein fusion strategy is still evolving and definitely requires further optimization.

Reply: We thank the reviewer for bringing this new point to the discussion. Short ssOligo DNA in conjunction of Cas9 were previously used for HDR in plants Shan *et al* 2013. Definitely due to phosphorothioated stability from nucleases, modified ssDNA would be available for longer time in the cells and hence have the chance to increase HDR. Here we would like to mention that application of our VirD2-Cas9 fusion approach along with RB containing phosphorothioated templates clearly enhanced HDR compared to phosphorothioated templates without RB. Our results clearly demonstrated the efficient role of the covalent tethering of the repair templates to the DSB via Cas9-VirD2 for targeted HDR in plants.